# The evolutionary impact of childhood cancer on the human gene pool

Ulrik Kristoffer Stoltze [1,2,3] ✉, Jon Foss-Skiftesvik [1,4],
Thomas van Overeem Hansen [2,5], Simon Rasmussen [6,7],
Konrad J. Karczewski [3,7,8,9], Karin A. W. Wadt [2,5] & Kjeld Schmiegelow [1,5] ✉

Germline pathogenic variants associated with increased childhood mortality must be subject to natural selection. Here, we analyze publicly available germline genetic metadata from 4,574 children with cancer [11 studies; 1,083 whole exome sequences (WES), 1,950 whole genome sequences (WGS), and 1,541 gene panel] and 141,456 adults [125,748 WES and 15,708 WGS]. We find that pediatric cancer predisposition syndrome (pCPS) genes [n = 85] are highly constrained, harboring only a quarter of the loss-of-function variants that would be expected. This strong indication of selective pressure on pCPS genes is found across multiple lines of germline genomics data from both pediatric and adult cohorts. For six genes [*ELP1*, *GPR161*, *VHL* and *SDHA/B/C*], a clear lack of mutational constraint calls the pediatric penetrance and/or severity of associated cancers into question. Conversely, out of 23 known pCPS genes associated with biallelic risk, two [9%, *DIS3L2* and *MSH2*] show significant constraint, indicating that they may monoallelically increase childhood cancer risk. In summary, we show that population genetic data provide empirical evidence that heritable childhood cancer leads to natural selection powerful enough to have significantly impacted the present-day gene pool.

With the rise of the genomic era, a growing number of childhood cancer patients are offered whole genome or exome sequencing, providing the ability to investigate every gene in the human genome at a previously unattainable scale. Yet, because every human carries numerous rare and even personal variants, knowing which genes to investigate and prioritize presents a substantial challenge.

Childhood cancers make up just one percent of all human malignancies[1] and have little semblance with the cancers seen in adulthood[2]. In children, cancer tends to (i) originate from stem and progenitor cells[3–9], (ii) have low mutational burden[2,10], (iii) have more oncogenic fusion genes[11–15], (iv) have greater epigenetic dysregulation[16–21], and (v) develop in apparent absence of environmental factors[22–26]. A sixth feature is that pediatric cancers, by definition, occur during childhood. Tragically, such cancers tend, especially historically, to cause death prior to reproduction[27]. Consequently, the transmission of associated childhood cancer risk variants is, in theory, under evolutionary pressure − an aspect that remains virtually unexplored.

According to the Darwinian theory of evolution[28], genetic alterations associated with a high risk of pre-reproductive fatal disease, such

[1]Department of Pediatrics and Adolescent Medicine, Rigshospitalet, Blegdamsvej 9, Copenhagen, The Capital Region, Denmark. [2]Department of Clinical Genetics, Rigshospitalet, Blegdamsvej 9, Copenhagen, The Capital Region, Denmark. [3]Program in Medical and Population Genetics, The Broad Institute of MIT and Harvard, Merkin Building, 415 Main St, Cambridge, MA 02142, USA. [4]Department of Neurosurgery, Rigshospitalet, Blegdamsvej 9, Copenhagen, The Capital Region, Denmark. [5]Department of Clinical Medicine, University of Copenhagen, Blegdamsvej 3B, Copenhagen, Denmark. [6]Novo Nordisk Foundation Center for Protein Research, University of Copenhagen, Blegdamsvej 3B, Copenhagen, Denmark. [7]The Novo Nordisk Foundation Center for Genomic Mechanisms of Disease, Broad Institute of MIT and Harvard, Cambridge, MA 02142, USA. [8]Center for Genomic Medicine, Massachusetts General Hospital, 55 Fruit St, Boston, MA 02114, USA. [9]Analytic and Translational Genetics Unit, Massachusetts General Hospital, 55 Fruit St, Boston, MA 02114, USA. ✉e-mail: ulrik.kristoffer.stoltze@regionh.dk; kjeld.schmiegelow@regionh.dk

as childhood cancer, will be subject to natural selection due to a reproductive disadvantage[29]. In contrast, 90% of adult cancers occur in individuals older than 50 years[27] at which age more than 99% of all offspring will have been born[30,31]. Naturally, germline variants causing cancer this late in life will have been passed on to the next generation by the time cancer (and related mortality) occurs - hence natural selection for the adult cancer-causing variants is likely much weaker.

Along with the recent revolution in DNA sequencing, several genes have been shown to be associated with a high risk of childhood cancer, and germline variants in these genes are relatively common in pediatric pan-cancer studies found in approximately 10% of cases, many being de novo[10,32–35]. If these changes tend to be selected out, human adults should not carry them. With the ongoing aggregation of large datasets of genetic data from adult humans[36,37] it is, for the first time, possible to empirically demonstrate whether this evolutionary theory of genetic childhood cancer risk is evident in the gathering human pangenome (Fig. 1A). Population germline genomic data show patterns of mutational constraint, i.e., some genes in the human gene pool harbor far fewer deleterious variants than is to be expected - an indicator that such *constrained* genes have been under selective pressure (Fig. 1B).

As a metric for evolutionary pressure, mutational constraint has been used to elucidate high-risk genetics in other tragic pediatric phenotypes, such as autism[38], stillbirth[39], and across rare diseases[40]. Nevertheless, the relevance of constraint metrics for cancer risk genes, particularly those with penetrance in childhood, remains unexplored. In a field replete with selection, survival, and ascertainment biases, constraint should improve the level of confidence in known childhood cancer risk genes, as well as aid scientific discovery by linking genes now known to be evolutionarily constrained with risk of childhood cancer. Deep exploration of this phenomenon could change how pediatric cancer risk is assessed and understood in a fundamental way.

In this work, we firstly, exposit the evolutionary theory of genetic childhood cancer risk based on the mutational constraint spectrum data presented by Karczewski et al[36]., and, secondly, employ these insights to review the genes and syndromes that deviate significantly from the theory.

## Results

### pCPS genes with monoallelic risk phenotypes show constraint

Using reported metrics of constraint[36], the 85 Category 1 pediatric cancer predisposition syndrome (pCPS) genes[41] (Supplementary Data 3, 4) show a mutational spectrum in the human gene pool markedly lower than observed in the remainder of the human exome (more than 19,000 genes). Specifically, pCPS genes show just 27% of the LoF mutations that would be expected (mean observed vs. expected number of LoF variants (LOE) ratio), a clear difference from all other genes in the human genome (Mann-Whitney $U$ test, mean pLoF observed vs. expected upper bound fraction (LOEUF) score 51% vs. 95%, $p = 6.379e-15$, Fig. 2A, B).

The 85 pCPS genes were either autosomal dominant (AD, $n = 59$), autosomal recessive (AR, $n = 23$), or X-linked recessive (XLR, $n = 3$) (Fig. 2C, Supplementary Data 4). Theoretically, strong signals of LoF constraint should only be observed in genes where a single LoF variant (monoallelic) confers an increased risk of reproductive disadvantage. Hence the AR pCPS genes and the nine pCPS genes (all AD) known to cause cancer predisposition through gain-of-function (GoF) variation were considered as separate groups (Fig. 2C, Supplementary Data 3). In keeping with this, 34 (64%) of the 53 AD(LoF)/XLR pCPS genes were *constrained*, while the same was true for just two (9%; *MSH2* & *DIS3L2*, discussed later) of the 23 AR pCPS genes and four (44%; *PIK3CA*, *BRAF*,

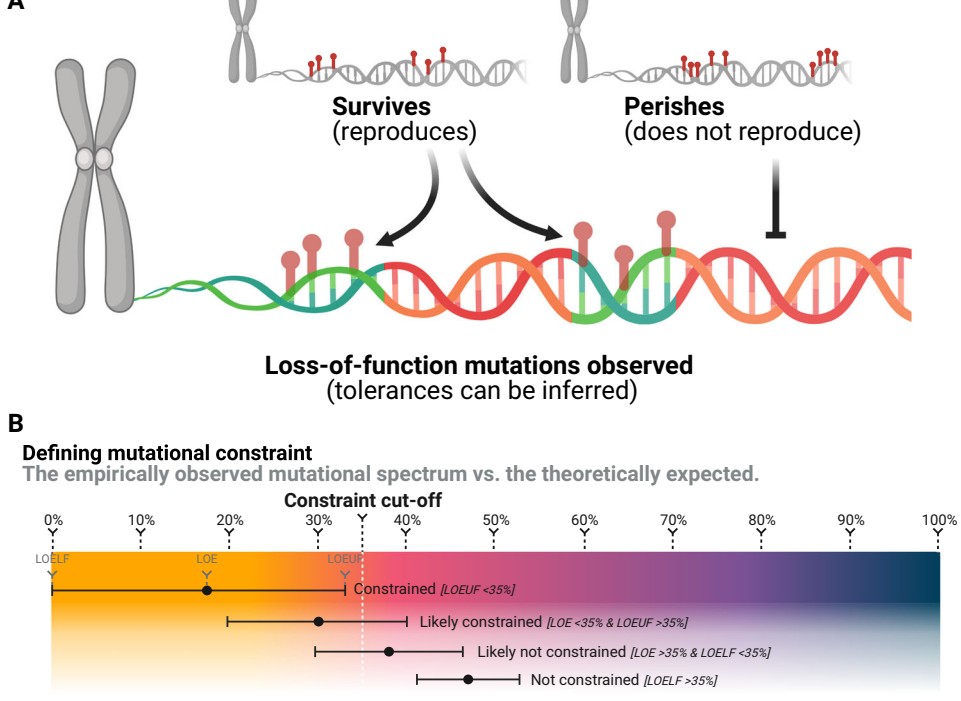

**Fig. 1 | Mutational survival bias generates constraint. A** Simplified illustration of survival bias of genomic variation; variants without reproduction-limiting effects survive, while variants limiting reproduction perish. This drives mutational constraint, which may be used for inferences of variant tolerances. Created with BioRender.com. **B** Purely illustrative examples of the definitions of mutational constraint used in the literature and/or in this study (not showing constraints of an actual gene). LOE refers to the specific ratio of observed vs. expected number of loss-of-function variants. LOELF and LOEUF, refer to the lower and upper bound fraction of the 90% confidence interval of the LOE score, respectively.

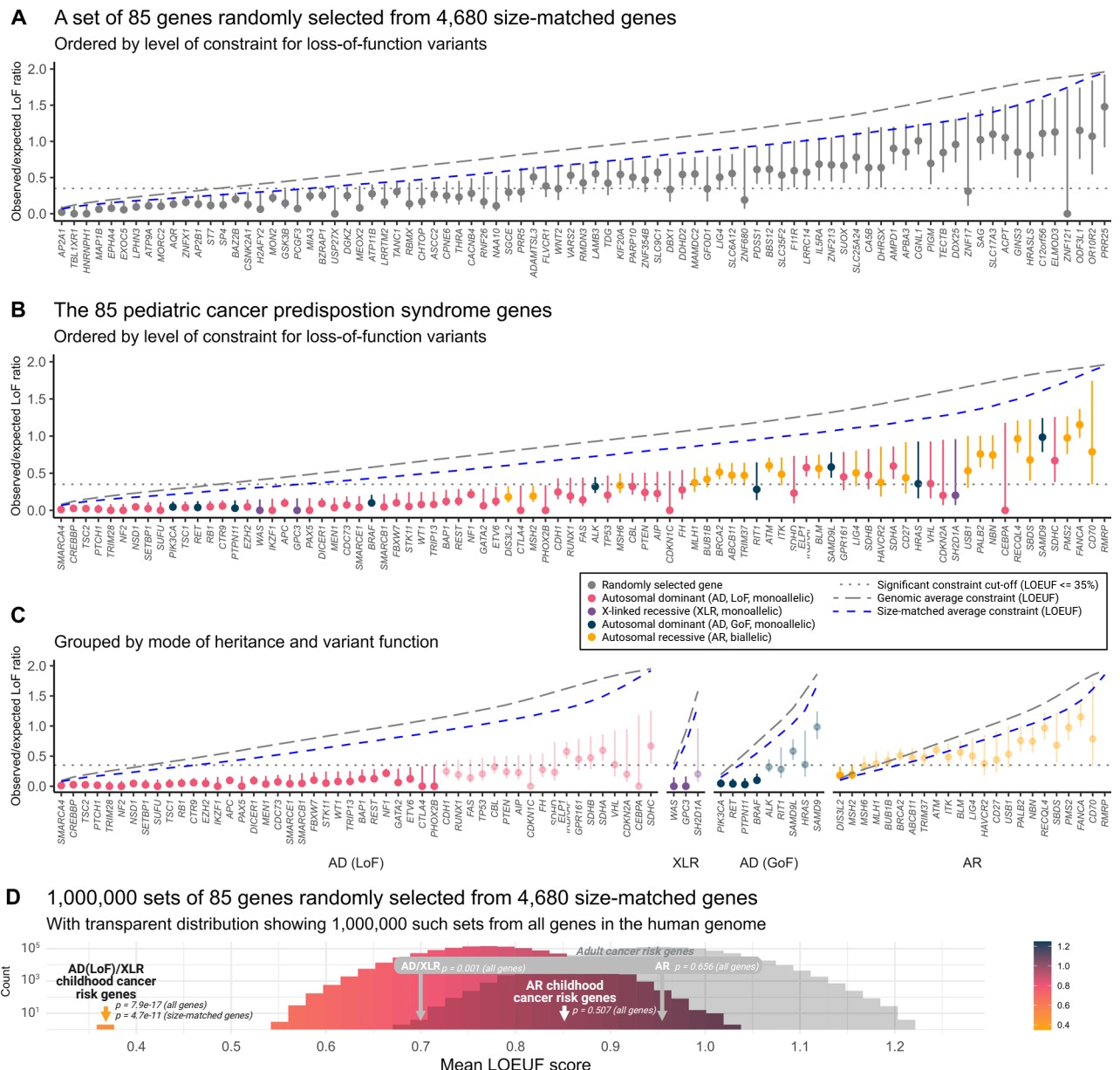

**Fig. 2 | Loss-of-function mutational constraint in genes linked with childhood cancer risk.** Panels (**A**–**C**): Observed/expected loss-of-function (LoF) mutation ratios for genes associated with a high risk of pediatric cancer. Data represent the LoF observed vs. expected (LOE) ratio, with error bars spanning the LoF observed vs. expected lower bound fraction (LOELF) to the LoF observed vs. expected upper bound fraction (LOEUF) corresponding to 90% confidence intervals ($n = 141,456$ individual adult humans). Constrained genes, with LOEUF score > 0.35, are indicated using faded colors in panel (**C**). Italicized x-axis labels refer to gene names. LoF constraint metrics could not be calculated for *RMRP* as it is a non-coding RNA gene. AD autosomal dominant (monoallelic), XLR X-linked recessive (monoallelic), AR autosomal recessive (biallelic). Panel **D**: A histogram showing mean LOEUF scores as observed in 1 million random samples of 85 size-matched human genes. The observed averages for both genes associated with pediatric-onset cancer predisposition syndromes and adult-onset cancer predisposition syndromes are shown subsetted for monoallelic genes (AD/XLR) and biallelic genes (AR). Color gradient corresponds to LOEUF score as indicated. *p*, the two-sided Mann-Whitney *U* test *p*-value for a comparison of the group versus all other or size-matched genes as indicated. Source data are provided as a Source Data file.

*RET* & *PTPN11*) of the nine AD(GoF) pCPS genes. The high rate of the latter group is likely explained by severe non-cancer phenotypes associated with monoallelic LoF variation in those same genes[42–44].

Genes with monoallelic LoF cancer risk phenotypes were more likely to be highly constrained than genes with biallelic risk phenotypes (Fisher's exact test, 34/53 vs. 2/23, OR = 18.1 [95% CI 3.8−175.4], *p* = 9.588e-6), while such a difference was not clear in comparison with AD(GoF) pCPS genes (Fisher's exact test, 34/53 vs. 4/9, OR = 2.2 [95% 0.4−12.5], *p* = 0.290). Unsurprisingly, pCPS genes associated with biallelic risk phenotypes did not show significantly higher constraint than the human exome in general (Mann-Whitney *U* test, mean LOEUF

score 85% vs. 95%, *p* = 0.507). In isolation, the 53 pCPS genes associated with monoallelic LoF risk phenotypes had a mean LOE ratio of just 14%, i.e., of the LoF mutations expected (Mann-Whitney *U* test, mean LOEUF score 37% vs. 95%, *p* = 7.909e-17).

The 53 AD(LoF)/XLR pCPS genes were significantly longer than all other human genes (Mann-Whitney *U* test, mean coding sequence 2436 bp vs. 1713 bp, *p* = 0.002); a factor which can impact confidence of constraint metrics[36]. To address this, we employed a subset of 4680 genes from the human genome matched by size (Mann-Whitney *U* test, mean coding sequence 2436 bp vs. 2483 bp, *p* = 0.237), against which AD(LoF)/XLR pCPS genes still showed significantly higher constraint

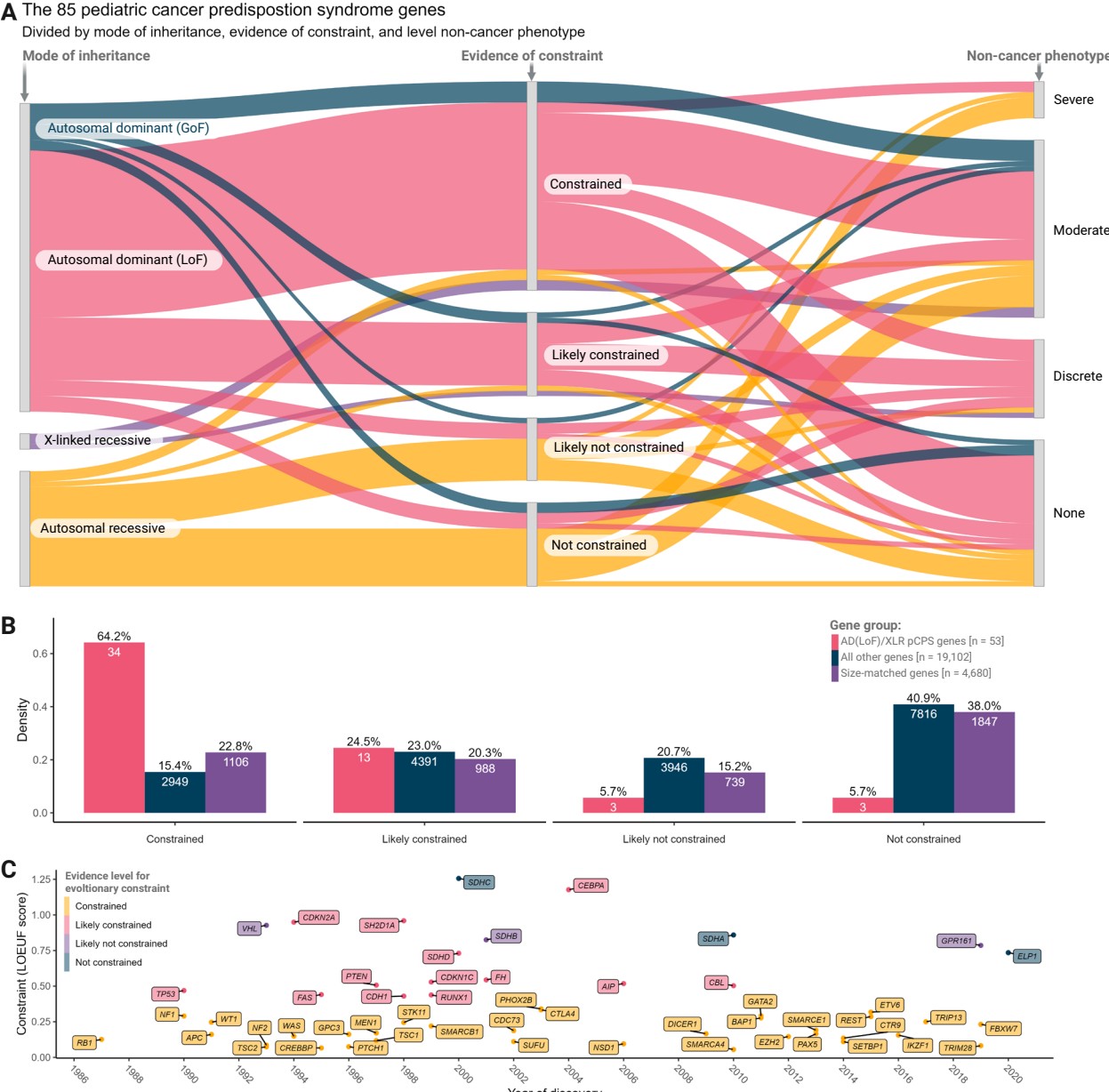

**Fig. 3 | Genic level of loss-of-function constraint in phenotypic context.**
**A** Sankey plot showing genes associated with, monoallelic and biallelic childhood cancer risk phenotypes and their relation to level of constraint for loss-of-function mutations and severity of associated non-cancer phenotype. Colors indicate mode of inheritance (MOI); autosomal dominant MOI caused by gain-of-function (GoF) variation (blue), autosomal dominant MOI caused by loss-of-function (LoF) variation (red), X-linked recessive MOI (purple), autosomal recessive MOI (yellow). **B** Bar plot showing density of genes with each level of constraint for each gene group as

indicated; autosomal dominant MOI caused by LoF variation or X-linked recessive MOI (AD(LoF)/XLR) pediatric cancer predisposition syndrome (pCPS) genes (red), all other genes (blue), and size-matched genes (purple). Numbers inside bars show number of genes. **C** Timeline showing year of pCPS gene discovery and level of constraint, with genes colored by level of constraint; constrained (yellow), likely constrained (red), likely not constrained (purple), and not constrained (blue). LOEUF, LoF observed vs. expected upper bound fraction. Source data are provided as a Source Data file.

(Mann-Whitney $U$ test, mean LOEUF score 37% vs. 77%, $p = 5.149\mathrm{e}{-11}$). In a repeated random sampling of 85 size-matched human genes from this pool of 4680 genes, no instance of equal or lower mean LOEUF score was seen ($n = 1,000,000$, mean LOEUF range 53–102%, Fig. 2A, D).

Among the 19 pCPS genes associated with monoallelic LoF risk phenotypes that had insufficient evidence to demonstrate the most significant level of constraint for LoF variants, 13 were *likely constrained*; three were *likely not constrained*; and three were *not constrained* (see Fig. 3A, B & Supplementary Table 1). The opposite was true for genes associated with biallelic risk phenotype; among the 21 such genes with

insufficient evidence to demonstrate definite constraint, just one was *likely constrained*; eight were *likely not constrained*; while 12 were *not constrained* (see Fig. 3A & Supplementary Table 1).

**pCPS genes with monoallelic cancer risk as the sole phenotype show constraint**
Around two-thirds of pCPS genes are also associated with phenotypes other than cancer (Fig. 3A, Supplementary Data 3). These non-cancer phenotypes can be severe, even fatal, and could thus in themselves lead to a gene becoming constrained. For example, Rubinstein-Taybi syndrome (caused by monoallelic variants in the highly constrained

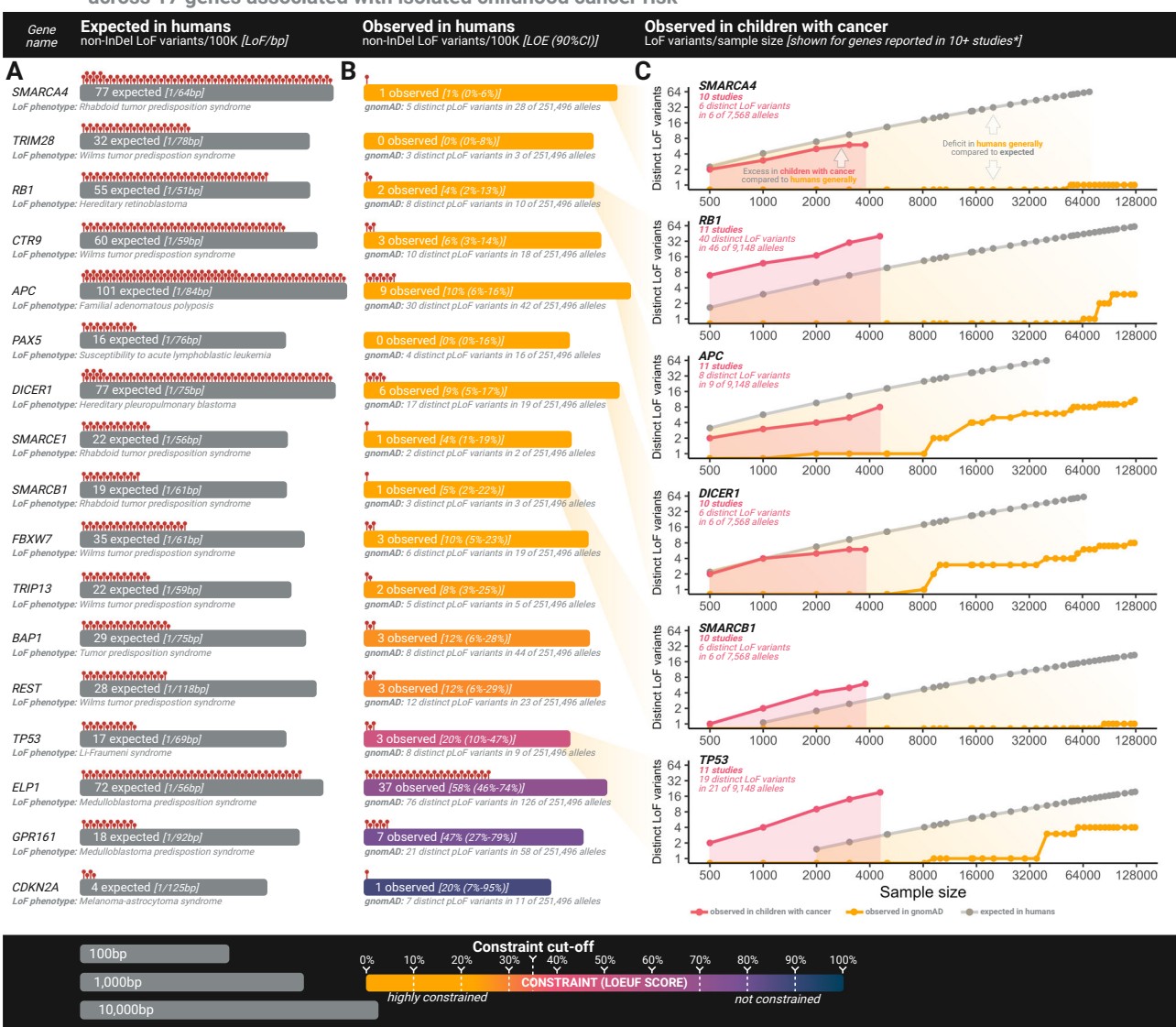

**Fig. 4 | Loss-of-function mutational spectra of genes linked with isolated childhood cancer risk. A** Visual illustration of the calculated number of non-insertion/deletion (non-InDel) loss-of-function (LoF) variants expected per 100,000 individuals in adults. Grey bar size represents gene size on a log10 scale as indicated. bp, base pair. **B** Number of non-InDel LoF variants actually observed in adults. Bar color represents level of constraint as indicated. **C** Log-scaled point graphs showing number of distinct LoF variants at various sample sizes, with colors representing gnomAD data, pediatric pan-cancer data or expected number as indicated. For gnomAD data and pediatric pan-cancer data, the dots with the highest *x*-axis value of each plot correspond to actually observed LoF variants in the full data, with other dots representing number of observed LoF variants at 38 downsampling steps (precomputed for gnomAD with pediatric pan-cancer internally computed to match, see "Methods and Data Availability"). The expected number of LoF variants at each downsampling step were precomputed as detailed by Karczewski et al.[36] and is publicly available from gnomAD (see Data Availability). *BAP1* and *CDKN2A* were reported in 10 pediatric pan-cancer studies, but were not included in this figure due to figure size/readability. Graphs for all 11 genes not shown here are in Supplementary Fig. 3. 100 K, 100,000 individuals, LOE, LoF observed vs. expected ratio, 90% CI, 90% confidence interval. Source data are provided as a Source Data file.

*CREBBP* gene) is associated with both congenital heart defects and with increased risk of childhood neuroblastoma and rhabdomyosarcoma[45]. Disentangling the true driver of the observed natural selection for this and similar genes, is not possible based on current data and understanding.

Hence, we looked at genes associated with monoallelic risk where increased risk of neoplasms is the prevailing known phenotype. The following 17 genes fulfill this criteria: *APC, BAP1, CDKN2A, CTR9, DICER1, ELP1, FBXW7, GPR161, PAX5, RB1, REST, SMARCA4, SMARCB1, SMARCE1, TP53, TRIM28, TRIP13* (see Figs. 3A, 4, and Supplementary Data 3). These genes remained significantly more constrained than both the average human and the average size-matched gene (Mann-Whitney

*U* tests; mean LOEUF score 31% vs. 95% & 76%, *p* = 1.378e-07 & 1.236e-05, respectively), while they had similar constraint to those AD/XLR pCPS genes that did have additional non-cancer phenotypes (Mann-Whitney *U* test; mean LOEUF score 31% vs. 42%, *p* = 0.356 [mean coding sequence, *p* = 0.297]). Thus, our analyses suggest that apparently isolated early life cancer risk can drive natural selection.

As further substantiation of this phenomenon, it is worthwhile comparing mutational variation in the observed human gene pool, not just to what is expected theoretically, but also to the mutational spectrum actually observed in children with cancer. Across 11 pediatric pan-cancer studies[10,33,34,46–53] covering 4,574 children with cancer, an approximate number of LoF variants per gene can be gleaned based on

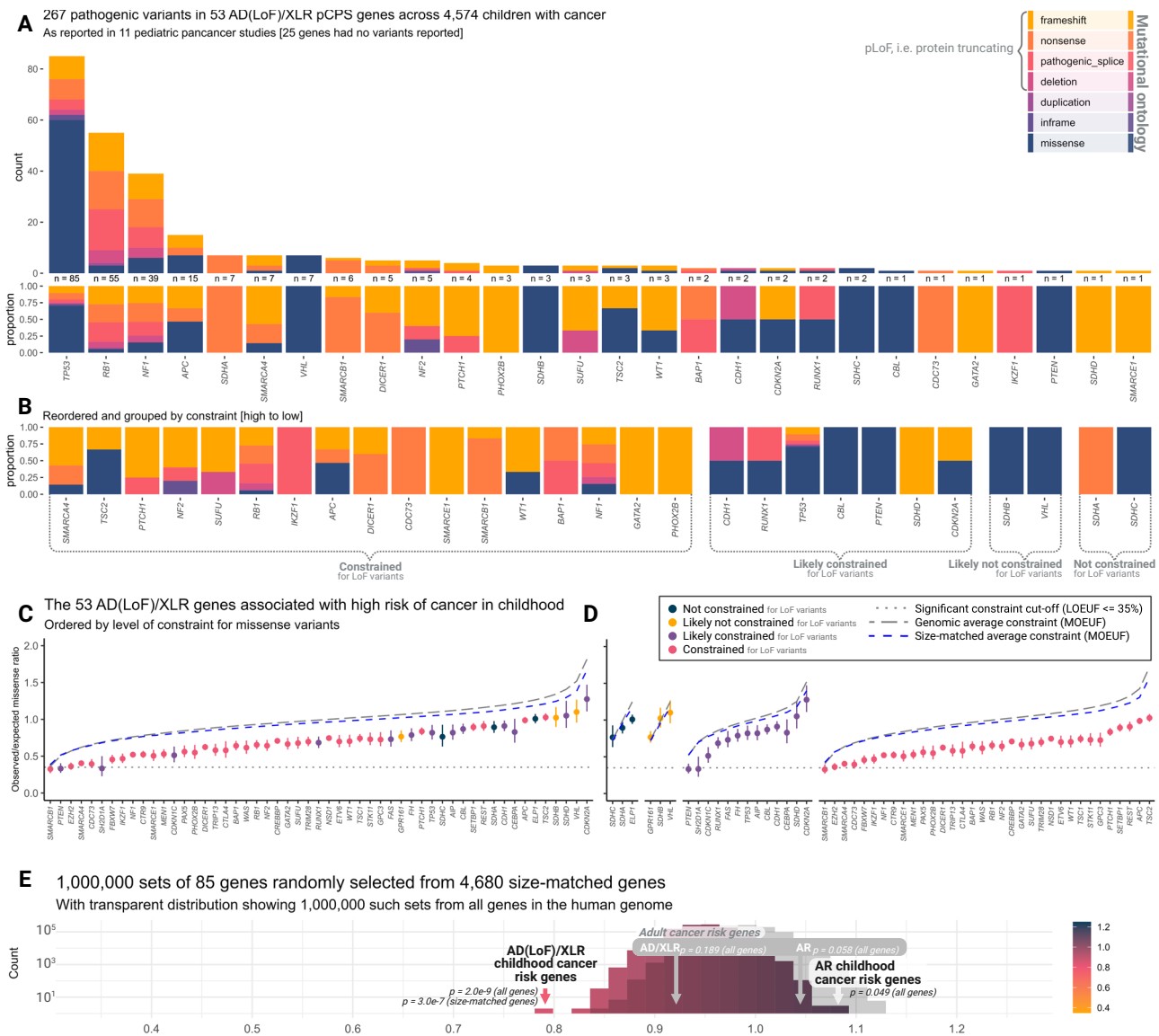

**Fig. 5 | Mutational spectra in children with cancer & missense mutational constraint. A** Bar plot showing mutational ontology (represented by color as indicated) of variants reported as pathogenic in 53 pediatric cancer predisposition syndrome (pCPS) genes with autosomal dominant mode of inheritance driven by loss-of-function or X-linked recessive mode of inheritance (AD(LoF)/XLR) genes as reported across 11 pediatric pan-cancer studies. Number of pathogenic variants reported and proportion of ontologies is given below each bar. Synonymous yet reportedly pathogenic variants (*n* = 1) were excluded. **B** Proportion of ontologies rearranged by level of constraint evidence. **C** Observed/expected missense variant ratio for 53 AD(LoF)/XLR pCPS. Dots represent the ratio, with error bars spanning the missense observed vs. expected lower bound fraction (MOELF) to the missense

observed vs. expected upper bound fraction (MOEUF) of a 90% confidence interval (based on exonic data from 141,456 individual adult humans). Italic *x*-axis labels refer to gene names. **D** As for panel (**C**), but grouped by each gene's level of constraint for loss-of-function variants (cf. Figs. 2, 3B). **E** Shows mean MOEUF scores as observed in 1 million random samples of 85 size-matched genes. The observed averages are shown for both genes associated with pediatric-onset cancer predisposition syndromes (pCPS) and adult-onset cancer predisposition syndromes, subsetted for monoallelic genes (AD/XLR) and biallelic genes (AR). *p*, the two-sided Mann-Whitney *U* test *p*-value for a comparison of the indicated group versus all other or size-matched genes as indicated. Source data are provided as a Source Data file.

the reported variants (Fig. 4 & 5A, Supplementary Data 5–26). For instance, *SMARCA4*, which increases risk of several childhood cancers[54], would, in theory, be expected to have 68 distinct non-insertion/deletion (non-InDel) LoF variants per 100,000 naturally, yet the adult human gene pool is severely constrained with just one such variant observed in gnomAD. Meanwhile, the *SMARCA4* gene pool for children with cancer (reported for 10 studies, *n* = 3784, Supplementary Data 16–26) identified six LoF variants. Downsampled, gnomAD data has no LoF variants at *n* = 3784. (Figs. 4, 5A, and Supplementary Fig. 3). The association between LoF variants in constrained genes and pediatric cancer suggests that such mutations would have had reduced

transmission between generations during human history. This, playing out over evolutionary time, is seemingly a partial driver of the constraint observable in the human pan-genome today.

## Some pCPS genes with monoallelic risk phenotypes are not constrained
The paragraphs above have focused on the constraint for LoF variants. Naturally, both missense variants as well as copy number variants may not lead to LoF but still cause CPSs. Indeed, the (likely) not constrained pCPS genes harbor several missense variants reported as pathogenic (Fig. 5B, C). Accordingly, we looked at evidence for constraint of

missense variants in AD/XLR pCPS genes (Fig. 5C). Yet, the sheer number of benign and inconsequential missense variants in the species-wide human genome means that the confidence of constraint metrics is drastically lower[36]. Hence, the negative predictive value is low, i.e., not finding constraint for missense variants does not provide meaningful confidence that constraint, perhaps in just specific exons or loci of a gene, is not present.

While no gene showed constraint according to the 35% cut-off used for LoF mutations, overall, the 53 AD(LoF)/XLR pCPS genes showed significant constraint for missense variants compared to the rest of the human exome (Mann-Whitney $U$ test; mean MOEUF score 79% vs. 99%, p = 2.045e-9) and size-matched genes (Mann-Whitney $U$ test; mean MOEUF score 79% vs. 95%, $p$ = 3.035e-7) (see Fig. 5C–E). In a new repeated random sampling of 85 size-matched genes, no instance of similar or lower mean MOEUF score was seen ($n$ = 1,000,000, mean MOEUF range 83–109%, Fig. 5E). Of note, constraint for missense mutation was only significant for those genes that were (likely) constrained for loss-of-function mutations, meaning that the AD(LoF)/XLR pCPS genes that were (likely) *not* constrained for pLoF mutations also showed no difference in missense mutation constraint (Mann-Whitney $U$ test; mean MOEUF score 99% vs. 99%, $p$ = 0.936) (see Fig. 5D).

### Relationship between selective pressure and constraint

As shown above, our analyses suggest that the risk of childhood cancer associated with a mutation (penetrance) could drive the signal of mutational constraint traceable in current data on the modern human pan-genome. In this regard, childhood cancer presents a compelling model for the overall concept, because historically, childhood cancer can reasonably be assumed to have been universally fatal. Therefore, even the modern childhood cancer penetrance associated with a given mutation may be considered equivalent to the historical negative selective pressure.

An obvious consideration becomes whether a relationship between the childhood cancer penetrance associated with each pCPS gene and level of constraint exists. However, reliable estimates of penetrance associated with most pCPS genes remain scarce. Still, for each of the AD(LoF)/XLR pCPS genes with some level of supporting evidence for penetrance [62%, 33/53, Supplementary Data 3], we categorized the childhood cancer penetrance as likely to be either very low/low (<0.1%/<1%) or high/very high (>1%, >5%). This revealed that pCPS genes reported to be associated with high to very high risk were more constrained than those with low to very low risk (mean LOEUF score 18% vs. 60%, multiple linear regression, $r$ = 0.58, $p$ = 1.100e-4 (corrected for gene size), Supplementary Data 3). In support of this, undoubtedly, somewhat biased observation, one may also consider de novo rates, i.e., how often a pCPS mutation is shown to be newly arisen in the carrying individual. Once more, reliable estimates are lacking, however, pCPS genes with some level of supporting evidence for de novo rate [60%, 32/53, Supplementary Data 3] were categorized as likely to be either very low/low (<1%/<10%) or high/very high (>10%, >20%). Again, pCPS genes reported as having high to very high de novo rates appeared to be more constrained than those with low to very low rates (mean LOEUF score 21% vs. 73%, multiple linear regression, r = 0.60, $p$ = 1.96e-05 (corrected for gene size), Supplementary Data 3). Just as residual confounders may exist, future, more reliable, data on penetrance and de novo rates may elucidate these relationships further.

### Pathogenic CPS gene mutations in children vs. adults

To further explore the theory presented here, a consideration of the germline pathogenic mutational landscape of adult pan-cancer is of interest. Among 10,389 adult individuals with 33 different types of cancer, a total of 8% had a pathogenic germline variant reported as pathogenic in any cancer-related gene[55]. Because several CPS genes are associated with both recessive (typically pediatric) and dominant (typically adult) phenotypes (e.g. CMMRD and Fanconi anemia), only

LoF mutations confidently reported as pathogenic, irrespective of zygosity or mode of inheritance should be considered. This resulted in just 229 mutations (2%) in adults versus 441 mutations (10%) in children with cancer across the 11 pediatric pan-cancer studies. Overall, the genes with reportedly pathogenic germline LoF mutations found in adults with cancer had an average of 85% of expected LoF mutations in the general human gene pool, versus 68% for the same in children with cancer (, mean LOEUF score 85% vs. 68%, multiple linear regression, $r$ = 0.14, $p$ = 1.29e-12 (corrected for gene size), Supplementary Fig. 1). This observation is in line with our previous findings that genes associated with adult-onset CPSs are significantly less constrained than genes associated with pediatric CPSs[56] (Supplementary Fig. 1). Importantly, both ascertainment, classification, and reporting biases, where currently there are no universal guidelines, influence this cross-investigational comparison. However, in combination with the other evidence presented, this merely adds to the overall picture of a phenomenon, which is highly likely to be grounded in biology.

Finally, we considered common genetic variants associated with cancer risk. Such variants typically show very low penetrance and generally confer only modest alteration in gene function, i.e. not loss-of-function. We compiled a list of single nucleotide polymorphisms (SNPs) associated with cancer risk based on review data from Sud et al.[57], and updated with newly discovered SNPs associated with childhood cancer from the GWAS catalog[58]. Across a total of 1047 genome-wide significant cancer risk SNPs associations, we analyzed the 722 distinct genes for which constraint metrics were calculated (673 genes in proximity to adult cancer risk SNPs and 49 genes in proximity to childhood cancer risk SNPs). Collectively, these genes were significantly more constrained for LoF variants than all other genes, yet, this difference was almost entirely driven by adult risk SNP genes (mean LOEUF score 80% [adult: 80%, childhood: 76%] vs. 96%, multiple linear regressions, $r$ = 0.14 [adult: 0.14, childhood: 0.14], $p$ = 1.58e-14 [adult: 7.44e-14, childhood: 0.067] (corrected for gene size, Supplementary Fig. 2). Of relevance to this study, the genes in proximity to adult and childhood cancer risk SNPs did not differ significantly (Mann-Whitney $U$ test, mean LOEUF score 80% vs. 76%, $p$ = 0.603), seemingly indicating that if the observed increase in constraint has any basis in biology, it is distinct from the phenomenon of pre-reproductive cancer incidence.

## Discussion

Here we have explored the theory that AD(LoF)/XLR genes associated with high risk of pediatric cancer show mutational constraint and we generally find strong evidence in its support. However, counter to the theory, six genes all believed to be associated with autosomal dominant pCPS driven by LoF genotypes, showed no constraint in the analyses above. This suggests that non-synonymous mutations (i.e. missense and LoF) in these six genes, may not have been under significant natural selective pressure. Interestingly, the genes clustered in two groups; (1) succinate dehydrogenase genes (three SHDx genes) and the *VHL* gene, and (2) *ELP1* and *GPR161*, both recently associated with SHH-activated medulloblastoma (MBSHH) susceptibility.

The succinate dehydrogenase family includes four genes listed on Byrjalsen et al.'s list[41] of pCPS genes (*SDHA*, *SDHB*, *SDHC*, and *SDHD*). These genes all cause the hereditary paraganglioma-pheochromocytoma syndrome[59], which has phenotypic overlap with von Hippel-Lindau syndrome (vHL, caused by pathogenic alterations in the *VHL* gene)[60]. None of these genes show a convincing level of constraint, calling into question either the frequency and/or the severity of childhood disease associated with these conditions. While pediatric-onset cancer in carriers with *SDHx* mutation is documented[61–64], low penetrance is clearly evident in epidemiological studies. The largest study of pathogenic SDHx mutations in humans found an approximate penetrance of just 20% by age 50 - and pediatric onset was rare (not enumerated in the study)[65,66]. The same study also

**Loss-of-function mutational spectrum in *ELP1* and *GPR161***
Mutations in the discovery cohorts of patients with medulloblastoma overlap with mutations found in gnomAD

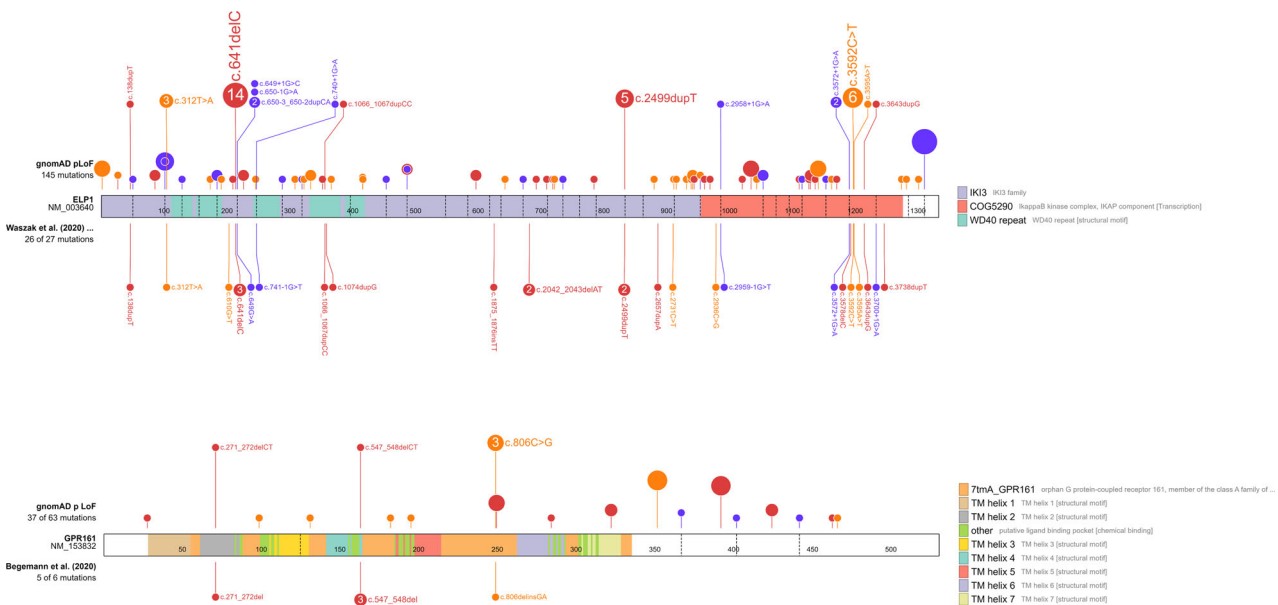

**Fig. 6 | Mutational spectra in recent medulloblastoma predisposition genes.**
Upper diagram shows the *ELP1* gene, with the upper tract representing the loss-of-function variants found in gnomAD v2.1 and the lower tract showing the loss-of-function variants found among patients with medulloblastoma by Waszak et al.[73]. Variants are expanded in the upper tract when they overlap with variants in the lower tract. Lower diagram shows the *GPR161* gene with the upper tract

representing the loss-of-function variants found in gnomAD v2.1 and the lower tract showing the loss-of-function variants found among patients with medulloblastoma. Variants are expanded in the upper tract when they overlap with variants in the lower tract. Variant color; orange, nonsense, purple, splice, red, frameshift. Source data are provided as a Source Data file.

showed that the mutational spectrum consisted of two thirds loss-of-function mutations. Collectively, these observations harmonize with the theory of constraint in childhood cancer risk genes, showing that in the vast majority of humans carrying SHDx pathogenic mutations, reproduction has seemingly not been limited by early disease and/or death. Intriguingly, the penetrance for variants in *SDHD* was twice as high as for other SDHx mutations, and more variants in *SDHD* (84%) were LoF. Seemingly in accordance with this, *SDHD* is the only succinate dehydrogenase gene that is *likely constrained* for LoF mutations (LOE ratio of less than 0.35), even though a phenotype is only present with paternal transmission[67].

Patients with pathogenic *VHL* mutations have nearly full penetrance of vHL by age 65 years[68], yet tumors are rare in childhood and are often benign[69–71]. Indeed, a study of survival in vHL patients going back as far as 1841 found no increased mortality in childhood[72]. Thus, both according to the evolutionary data presented here and clinical observations, pathogenic *VHL* mutations seemingly have only a limited impact on the carrier's reproductive potential.

Waszak et al[73]. recently reported that the *ELP1* gene, also known as *IKBKAP*, is associated with risk of MBSHH. The study found germline LoF mutations in 29 of 202 (14%) pediatric MBSHH cases, and concluded that *ELP1* is the most common MB predisposition gene (Supplementary Data 28). The observation was supported by a strong tendency for carriers to have the molecular SHHα subtype all showing in trans loss of chromosome 9q (loss of heterozygosity)[73]. In one of the 29 families, the germline *ELP1* LoF mutation was shown to segregate with the father who also had pediatric MB.

Despite this striking evidence, gene pool data shows that the *ELP1* gene is *not constrained* for pLoF mutations. In other words, while a greater number of children with MB have germline *ELP1* pLoF mutations, adults do not have a lower number of variants than would be expected. Other genes known to predispose to childhood MBSHH (such as *SUFU*, *PTCH1* and *TP53*) are all *(likely) constrained*, making the

apparent lack of constraint in *ELP1* notable as it conflicts with the idea that pLoF mutations in *ELP1* should confer high risk of childhood MB.

Of the 26 pLoF SNVs reported by Waszak et al.[73], 15 (58%) were also seen in healthy adults, including the c.168dupT variant shown to segregate with disease (Fig. 6 and Supplementary Data 27, 28). The most likely explanation for the lack of contraint in *ELP1* is likely to be low penetrance. Yet, since constraint is driven by a reproductive disadvantage, another possible explanation may be that individuals with childhood medulloblastoma (3 to 20 years of age at diagnosis) survived long enough to pass on the susceptibility mutation. However, historical data shows that even with the treatments available in the 1940s childhood MB led to death within a few years in nearly all cases[74]. Lastly, it is possible that a negative selective pressure is balanced out by positive selective pressure, which could, for instance, be driven by increased fertility. Similarly speculative theories have been suggested for other CPS genes[75].

In a very similar situation, Begemann et al.[76], reporting on the same data as Waszak et al.[73], found that six patients with infant-onset MB carried germline variants in the *GPR161* gene (five were pLoF)[76]. Here too, the MB subtype was universally MBSSH, and the tumors invariably showed loss of heterozygosity (chromosome 1q loss). At least one family member was tested in two of the six families, and in both cases the variant was inherited from a parent without personal or family history of MB. As discussed by Begemann et al.[76], all six (100%) of the identified mutations were also seen in healthy adults (Fig. 6 and Supplementary Data 29, 30). Hence, the argumentation mirrors that of *ELP1* above, and again low penetrance appears to be a likely explanation for the lack of evidence of pLoF constraint.

These observations could have important implications for the clinical utility of germline testing of these genes in MB patients and their family members. Here, any surveillance or family planning intervention should probably be approached with great caution, as the absolute risk of MB may only be moderate, likely somewhere between

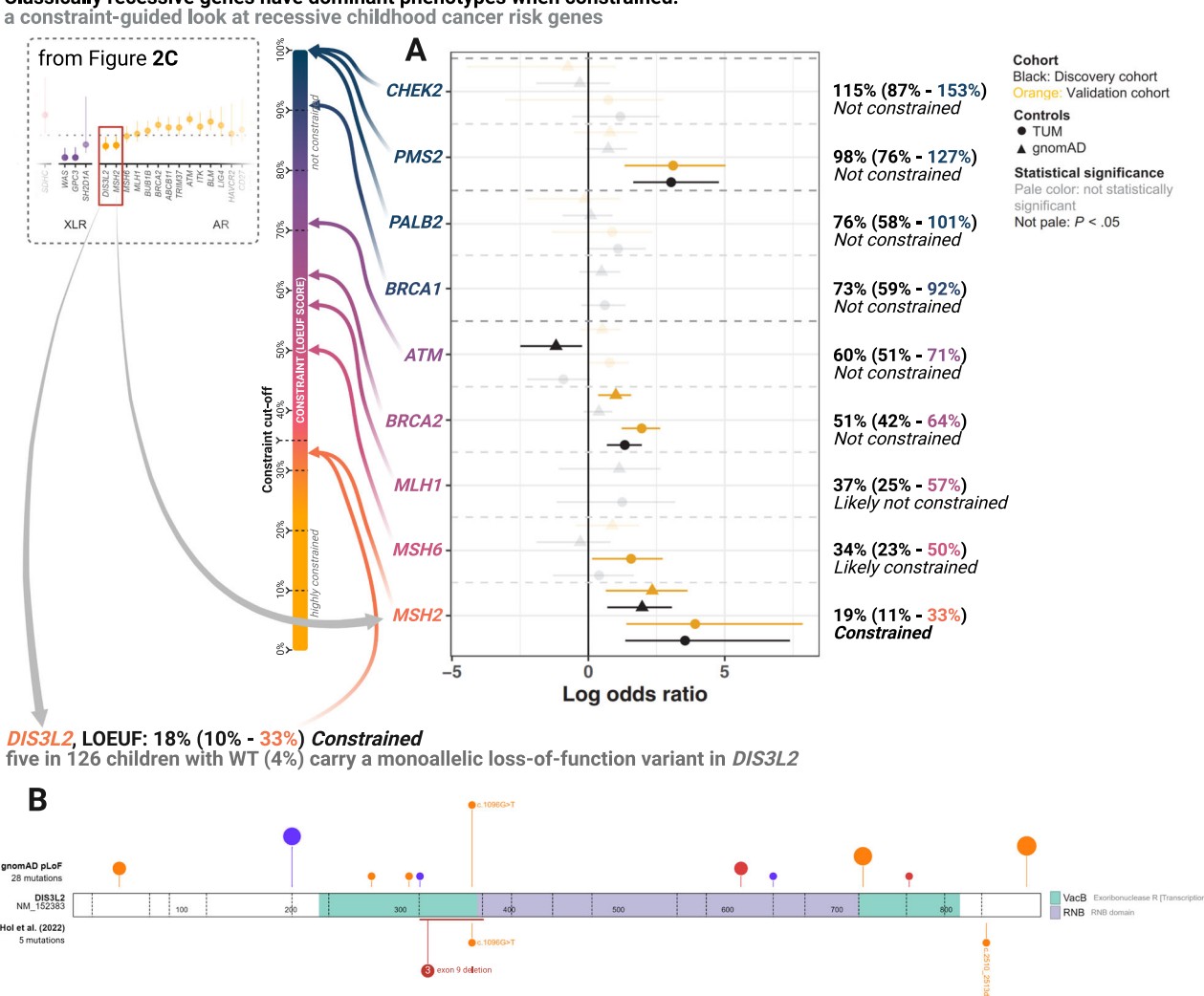

**Fig. 7 | Mutational constraint of genes typically linked with adult cancer (or, biallelically, childhood cancer) risk: shows data regarding the phenotypes associated with monoallelic loss-of-function of classically recessive childhood cancer risk genes. A** Modified adaptation from Kratz et al. (2022)[85] with permissions provided under the CC BY-NC 4.0 license showing the odds ratio of excess carriers of pathogenic variants (error bars indicate 95% confidence intervals) in children with pan-cancer (discovery cohort; *n* = 3775 & validation cohort; *n* = 1664) vs. healthy adults (gnomAD; *n* = 74,023 & TUM; *n* = 27,501) for nine classically adult cancer predisposition genes. "Gene-based burden testing was performed and odds ratios, 95% confidence intervals, and *P* values were calculated using the 2-sided Fisher exact test"[85]. Genes are colored and rearranged according to their loss-of-function observed vs. expected upper bound fraction (LOEUF) score as indicated by the central constraint spectrum bar. On the right exact constraint metrics; loss-of-function observed vs. expected ratio, loss-of-function observed vs. expected lower bound fraction score and LOEUF score, as well as constraint level, are listed. TUM refers to a control cohort of cancer-free individuals sequenced at Technical University of Munich. **B** The upper tract shows the *DIS3L2* loss-of-function variants found in gnomAD v2.1. The lower tract shows the loss-of-function variants found among patients with Wilms tumor (WT) by Hol et al.[82]. Variants are expanded in the upper tract when they overlap with variants in the lower tract. Variant color; orange, nonsense, purple, splice, red, frameshift. Source data are provided as a Source Data file.

---

ordinary CPSs and cancer susceptibility SNPs that mostly have odds ratios between one and two, and generally have no clinical impact.

Of note, recent work related to germline *ELP1* and *GPR161* pLoF variants indicated that the associated cancer risk may be as low as 1 in 430 and 1 in 2500, respectively[77]. It seems that such low penetrance may not drive a gene to become significantly constrained.

Conversely to the discussion above, two of the 23 genes associated with pCPSs that have an AR mode of inheritance show significant constraint [*DIS3L2* & *MSH2*,] (Fig. 2C), suggesting that these two genes have a dominant phenotype that is severe enough to limit a carrier's chance to pass on their genes.

Perlman syndrome is a severe AR disorder long-known to be caused by pathogenic variants in *DIS3L2*, and associated with high risk of Wilms tumor (WT) in infancy or early childhood[78]. In line with the

gene's significant level of constraint, heterozygous *DIS3L2* variants have been linked to WT development in case reports[79–81]. Recently, this was corroborated in a unselected prospective study of WT which found that 5 out of 126 children with WT (4%) carry monoallelic germline LoF variants in the gene[82] (Fig. 7B). Across the 11 pediatric pan-cancer studies, *DIS3L2* was on the analysis panel for 2,786 patients (Supplementary Data 16–26), reporting one LoF variant[83] in a child with WT (Supplementary Data 5–15). However, many studies have not reported heterozygous carrier states for genes presumed to have AR mode of inheritance, and thus a reporting bias in these studies, which all predated the paper suggesting the heterozygous link to Wilms tumor risk[82], likely exists.

The *MSH2* gene is part of the mismatch repair (MMR) gene family. The AR phenotype associated with pathogenic mutations in the MMR

genes is termed constitutional mismatch repair deficiency (CMMRD) and considered the single most penetrant pCPS. Lynch syndrome (LS) is the AD phenotype associated with the same genes and is predominately an adult-onset CPS. In LS, *MSH2* variants account for the majority of cases and they are associated with earlier cancer onset than observed for other MMR genes[84]. This variability in penetrance is in line with *MSH2* showing significant constraint, while *MSH6*, *MLH1*, and *PMS2*, all associated with later onset and/or lower penetrance, do not. Of note, biallelic *PMS2* variants account for the majority of cases of the biallelic phenotype, CMMRD, and the gene shows clear lack of constraint. Remarkably, a recent study by Kratz et al[85], investigating pediatric cancer risk associated with pathogenic variants in nine genes classically associated with adult-onset CPSs, found that only *MSH2* was consistently associated with childhood cancer risk[85]. Once again, this matches the constraint metrics observed for the nine genes investigated, of which only *MSH2* was *constrained* (Fig. 7A). In a comparison, partly overlapping with the one conducted by Kratz et al., further empirical support is seen; of 4,574 children with cancer tested for *MSH2* seven (0.15%, 7/4,574) carried a LoF variant in the gene (Supplementary Data 5−26), compared to 31 such carriers among 125,564 individuals in gnomAD (0.02%) (Fisher's exact test, 7/4,574 vs. 31/125,564, OR = 6.2 [95% CI 2.3−14.3], p = 3.219e-4, Supplementary Data 34). However, and importantly, this observation has some data overlap with the original paper describing an excess burden of *MSH2* variants in children with cancer[85]. Other studies have made observations substantiating pediatric-onset cancer predisposition for MMR genes, yet these have generally lacked controls[86,87]. Among the 5 heterozygous *DIS3L2* LoF variants seen in the 126 children with WT mentioned above, one overlapped with the 34 reported in gnomAD (Fisher's exact test, 5/126 vs. 34/124,700, OR = 151.3 [95% CI 45.5−397.6], p = 5.41e-10, Supplementary Data 35). Unlike *ELP1* and *GPR161*, for both the *MSH2* and *DIS3L2* variants seen in children with cancer, only a single variant overlapped with those found in the background population, which could mean that LoF variants associated with childhood cancer have a distinct spectrum, although this requires further data.

Broadly, of the nearly 3,000 human genes that show significant constraint for LoF variants, the majority are not (yet) linked to human disease or otherwise explained[36]. It is noteworthy that several of the genes associated with high risk of childhood cancer discovered in recent years were among the constrained genes, for example: *ETV6*, discovered in 2015[88−91], LOEUF = 0.318; *FBXW7*, discovered in 2019[92], LOEUF = 0.232; and, as discussed above, *DIS3L2*, discovered in 2022[82], LOEUF = 0.325 (Fig. 3C).

Still, constraint is typically not discussed in the discovery of pCPS genes, a practice we hope to change with the insights gathered here. Furthermore, it seems logical that analyses focused on constrained genes have the potential to accelerate discovery of genes specifically associated with pediatric-onset, as opposed to adult-onset, cancer. We have recently applied this approach in childhood cancer cohorts with germline whole genome sequencing available, and identified candidate pCPS genes[56,93]. Ultimately, due to the clinical implications of diagnosing CPSs, great difidence must be exercised in drawing causal genotype-phenotype conclusions. Here constraint metrics cannot stand alone, but must compound with other observations. Yet, for AD(LoF)/XLR genes, any clear lack of constraint should be taken as an indicator that the gene may not confer high risk of cancer in childhood.

In conclusion, multiple sources of germline genomic data strongly support that genetic pediatric cancer risk is subject to higher natural selection and have driven constraint in numerous genes. Accordingly, investigators of genetic etiology of childhood cancer should analyze and report rare variants identified in constrained genes and their associated phenotypes, as this can promote discoveries of CPS and improved understanding of high-penetrance genetic childhood cancer risk.

## Methods

In order to ensure reproducibility the full, commented code used for data wrangling, analysis, and visualization have been uploaded to GitHub (see Code Availability in main text). The primary code was divided under headers and named chunks, which are referenced when relevant below.

### Empirical data on the variation of the human pangenome

Single nucleotide variant metadata from 141,456 adult humans published by Karczewski et al. in 2020[36], including their calculations of mutational constraint, was incorporated in this study by downloading their Supplementary Dataset 11 (publicly available as open-access). Only canonical gene transcripts were used. In this data, specific metrics on both empirically observed variants and theoretically expected variants are available. Predicted loss-of-function (pLoF) variants observed vs. expected (LOE) scores refer to the specific ratio of observed vs. expected number of pLoF variants. pLoF observed vs. expected upper/lower bound fraction (LOEUF/LOELF) scores refer to the lower and upper bounds of the 90% confidence interval of the LOE score, respectively. In keeping with the literature[36,94], a gene was designated as *constrained* when it showed a LOEUF score lower than 0.35 based on random distribution across the genome. For the purposes of our study, genes with pLoF observed vs. expected (LOE) ratio under 0.35 were designated *likely constrained*; genes with pLoF observed vs. expected (LOE) ratio over 0.35 were designated *likely not constrained*; and genes with an observed vs. expected lower bound fraction (LOELF) score higher than 0.35 were designated *not constrained* (Fig. 1B). In essence, *not constrained* refers to genes where the current data does not support that the gene is, or is likely to be, *constrained* using the 35% LOEUF cut-off. Throughout, higher constraint refers to lower LOEUF score.

GnomAD v2.1 data was also implemented in analyses using publicly available variant-resolution data from the 125,748 exomes ("All chromosomes sites VCF", https://storage.googleapis.com/gcp-public-data--gnomad/release/2.1.1/vcf/exomes/gnomad.exomes.r2.1.1.sites.vcf.bgz, see primary code, header: Fig. 4 (incl. gnomad data import), chunk: gnomad_data_load_and_coverage_filter). This data was subsetted using filter (pLoF == "HC") on a separate high-performance compute cluster, extracting only variants only considered high confidence by LOFTEE (https://github.com/konradjk/loftee).

### Size-matched gene set

In order to account for gene size (included as a variable, coding sequence (CDS) lengths, in Supplementary Dataset 11 by Karczewski et al.[36]) we arranged pCPS genes according to gene size and grouped genes into nine subsets of equal size. Using the size ranges from these nine groups of pCPS genes, we randomly sampled all genes (not excluding pCPS genes), to create the maximum size-matched subset, satisfying the criteria that (1) genes appeared only once and (2) each group contained an equal number of genes (see primary code, header: Data cleaning, annotation and preparation, chunk: sizematching). The resultant group of size-matched genes was compared to the genes of interest to ensure that the two groups were no longer significantly different.

### Selecting cancer predisposition syndrome genes with pediatric-onset

The paper *Selection criteria for assembling a pediatric cancer predisposition syndrome gene panel* (2021) by Byrjalsen et al.[41] lists a total of 85 genes deemed *Category 1* for childhood cancer risk (Supplementary Data 3); i.e. a pathogenic variant (i) in a minimum of five children (aged 0−18 at diagnosis) with (ii) a cancer diagnosis in at least two independent families. These minimum requirements are independent of the prevalence in adult-onset cancer patients[41].

In our work, these 85 genes are referred to as the pediatric-onset cancer predisposition syndrome (pCPS) genes. Throughout, we distinguish between biallelic phenotypes, caused by two pathogenic variants sitting in trans in a gene (i.e. autosomal recessive), and monoallelic phenotypes, caused by one single variant in a gene (i.e. autosomal dominant and X-linked recessive), as well as whether the associated pediatric phenotype was driven by loss-of-function or gain-of-function genotypes. (Supplementary Data 4).

A separate set of cancer predisposition syndrome (CPS) genes were designated as adult-onset cancer predisposition syndrome (aCPS) genes using the following methodology, also employed in our previous study[56]. Using a list of all CPS genes published by Rahman (2014)[95], we filtered out all gene/mode-of-inheritance pairs that were designated as associated with pCPS (see above). The remaining 62 genes/mode-of-inheritance combinations were designated as aCPS genes (Supplementary Data 4, see primary code, header: Data cleaning, annotation and preparation, chunk: annotating_constraint).

## Combined germline mutational spectrum in patients with childhood cancer

All pediatric pan-cancer studies, published in June 2022 or earlier and including germline sequencing of 100 or more participants were included irrespective of the sequencing strategy (targeted, WES, or WGS). First, the data were loaded into R in their original published format; the 'rbind()' function was used to combine data from the following studies: Zhang et al. (2015)[33], Parsons et al. (2016)[46], Mody et al. (2016)[47], Oberg et al. (2016)[48], Gröbner et al. (2018)[10], Wong et al. (2020)[49], Byrjalsen et al. (2020)[34], Fiala et al. (2021)[50], Newman et al. (2021)[51], Stedingk et al. (2021)[52], and Wagener et al. (2021)[53] (Supplementary Data 5–15). Similarly, gene panels from each study were combined using 'rbind()' (Supplementary Data 16–26). A patient and data overlap between Zhang et al. (2015)[33] and Gröbner et al. (2018)[10] was noted by Gröbner et al. (2018);[10] "[out of] $n = 914$ individual patients, about 25% of samples overlapping with the previous study". While not stated explicitly in the publication, it was assumed that this refers to the 259 samples [28.4%, 259/914] that shared a St. Jude ID (format; SJxxxnnn). Hence, this overlap was accounted for by filtering out patient IDs from Zhang et al. (2015)[33] that also appeared in Gröbner et al. (2018)[10] (see primary code, header: Data cleaning, annotation and preparation, chunk: metadata_homogen).

Next, nomenclature cleaning was performed on the mutational ontology, pathogenicity classification, and diagnosis type categories using the 'mutate()' function from the 'dplyr' package in R. As an example, likely pathogenic variants, reported as 'PP', 'likely pathogenic', 'C4: Likely pathogenic', 'Likely Pathogenic', 'Likely pathogenic', and 'LP' were homogenized to simply 'LP' (full details for all 300+ renamings are in primary code, header: Data cleaning, annotation and preparation, chunk: metadata_homogen). All reported variants were retained and no variants were filtered out, with the exception of the c.3920 T > A (p.Ile1307Lys) allele in the *APC* gene, that while reported as pathogenic in one[50] of the 11 studies, is considered a risk factor for cancer rather than a precipitator of the CPS associated with the gene. The c.3920 T > A (p.Ile1307Lys) allele has not been shown to cause childhood cancer (see primary code, header: Fig. 5, chunk: pancan_plots).

Any germline variant reported as either likely pathogenic or pathogenic may be referred to simply as pathogenic in the manuscript text. Variants were considered pLoF if the variant ontology was frameshift, nonsense or splice donor/acceptor regardless of pathogenicity. To make variation observed in children with cancer comparable to those seen in gnomAD, we used pre-computed downsamples of the gnomAD data (see under header 'Data Availability' in the main manuscript text; see primary code, header: Fig. 4 (incl. gnomad data import), chunk: downsampling_import) filtered to variation affecting canonical transcripts. Importing this data (https://storage.googleapis.com/gcp-publicdata--gnomad/release/2.1.1/constraint/gnomad.v2.1.1.lof_metrics. downsamplings.txt.bgz), we generated plots comparing the observed number of LoF variation in the two groups by downsampling the combined germline data reports from pediatric pancancer studies (downsampling was done, as allowed by sample size, to the same levels as those precomputed in the gnomAD data). A total of 38 possible downsampling steps were used [10, 20, 50, 100, 200, 500, 1000, 2000, 3070, 5000, 5040, 8128, 9197, 10000, 10824, 15000, 15308, 17296, 20000, 25000, 30000, 35000, 40000, 45000, 50000, 55000, 56885, 60000, 65000, 70000, 75000, 80000, 85000, 90000, 95000, 100000, 110000, 120000]. In the gnomAD dataset, at each of these downsampling steps the observed number of distinct LoF variants were pre-computed, as were the expected number of distinct LoF variants, calculated using the same methods employed for the gnomAD-wide calculations, as detailed by Karczewski et al.[36]. In our downsampling of the pediatric pancancer cohort data, we employed an iterative for-loop (see primary code, header: Fig. 4 (incl. gnomad data import), chunk: downsampling_for_pancan). This meant that the for-loop began by counting all distinct LoF variants (as defined below) across all available studies (variable by number of studies with the given gene included on panel (see Supplementary Data 16–26)). For genes included in all 11 pediatric pancancer studies, the total number of childhood cancer cases with data was 4,574 (9,148 alleles). Each unique patient was labeled either with their specific variant or as having no variant detected. From this set, using the sample_n function in the R package dplyr, a random sample, corresponding to the nearest lower downsample step ($n = 3,070$ in the example), was extracted. LoF variants in the extracted random sample were counted, and then, a random sample corresponding to the next downsample step ($n = 2,000$ in the example) was taken from the previous step ($n = 3,070$ in the example). This process ran iteratively through all downsampling steps (9 steps [10, 20, 50, 100, 200, 500, 1000, 2000, 3070] in the example).

## Statistical testing

We used R version 4.0.2 (2020-06-22) for all statistical tests (see primary code, header: All statistical tests, chunk: all_stat_tests). All reported P values are two-sided. No data samples were analyzed more than once. Confidence parameters are reported in-text for each test.

## Inclusion and ethics

This study did not include original data and, as such, neither qualified for nor required ethical oversight.

## Software and tools

External data was downloaded in .xlsx or .csv format and compiled in sheets, without edits, in a Microsoft Excel workbook. All subsequent data wrangling, analysis, calculations and some visualizations were done using R (version 4.2.2) through RStudio (version 2022.07.1). Non-computational illustrations were made using the browser-based graphical tool BioRender. Illustrations of exon/protein and variants were made using the browser-based graphical tool ProteinPaint[96] on the PeCan platform. Scripts were written in Rmarkdown and are available on github (see Code Availability in main text). The scripts are preambled by the required packages, which, by *name(version number)*, are; readxl(1.4.1), backports(1.4.1), systemfonts(1.0.4), plyr(1.8.8), lazyeval(0.2.2), splines(4.2.2), TH.data(1.1-1), digest(0.6.31), htmltools(0.5.4), fansi(1.0.3), memoise(2.0.1), googlesheets4(1.0.1), tzdb(0.3.0), penxlsx(4.2.5.1), remotes(2.4.2), modelr(0.1.10), matrix-Stats(0.63.0), sandwich(3.0-2), prettyunits(1.1.1), colorspace(2.0-3), vest(1.0.3), textshaping(0.3.6), haven(2.5.1), xfun(0.36), callr(3.7.3), crayon(1.5.2), jsonlite(1.8.4), libcoin(1.0-9), Exact(3.2,) survival(3.4-0), glue(1.6.2), polyclip(1.10-4), gtable(0.3.1), gargle(1.2.1), pkgbuild(1.4.0), clipr(0.8.0), Quandl(2.11.0), mvtnorm(1.1-3), DBI(1.1.3), miniUI(0.1.1).1, Rcpp(1.0.9), xtable(1.8-4), gridtext(0.1.5), foreign(0.8-83), bit(4.0.5), proxy(0.4-27), stats4(4.2.2), profvis(0.3.7), htmlwidgets(1.6.0), httr(1.4.4), modeltools(0.2-23), ellipsis(0.3.2), farver(2.1.1),

urlchecker(1.0.1), pkgconfig(2.0.3), ultcompView(0.1-8), dbplyr(2.2.1), utf8(1.2.2), labeling(0.4.2), tidyselect(1.2.0), rlang(1.0.6), later(1.3.0), munsell(0.5.0), cellranger(1.1.0), tools(4.2.2), cachem(1.0.6), generics(0.1.3), broom(1.0.2), evaluate(0.19,) fastmap(1.1.0), ragg(1.2.4), yaml(2.3.6), processx(3.8.0), bit64(4.0.5), fs(1.5.2), zip(2.2.2), coin(1.4-2), rootSolve(1.8.2).3, mime(0.12,) xml2(1.3.3), compiler(4.2.2), rstudioapi(0.14,) curl(4.3.3), e1071(1.7-12), gt(0.8.0), reprex(2.0.2), tweenr(2.0.2), broom.helpers(1.11.0), DescTools(0.99.47), stringi(1.7.8), ps(1.7.2), lattice(0.20-45), Matrix(1.5-1), vctrs(0.5.1), pillar(1.8.1), lifecycle(1.0.3), lmtest(0.9-40), ata.table(1.14.6), lmom(2.9,) httpuv(1.6.7), R6(2.5.1), promises(1.2.0).1, gld(2.6.6), sessioninfo(1.2.2), codetools(0.2-18), boot(1.3-28), MASS(7.3-58.1), assertthat(0.2.1), pkgload(1.3.2), withr(2.5.0), nortest(1.0-4), multcomp(1.4-20), expm(0.999-6), parallel(4.2.2), hms(1.1.2), quadprog(1.5-8), class(7.3-20), rmarkdown(2.19,) googledrive(2.0.0), shiny(1.7.4).

## Reporting summary
Further information on research design is available in the Nature Portfolio Reporting Summary linked to this article.

## Data availability
The raw data used in this study are fully available without any restrictions, either in the supplementary datasets or via the links provided both below and in the study code (see Code availability). Generated in this study are provided in the Supplementary Information/ Source Data file. Exceptions are GnomAD v2.1 exome data ("All chromosomes sites VCF", https://storage.googleapis.com/gcp-public-data--gnomad/release/2.1.1/vcf/exomes/gnomad.exomes.r2.1.1.sites.vcf.bgz); associated coverage data available from https://storage.googleapis.com/gcp-public-data--gnomad/release/2.1/coverage/exomes/gnomad.exomes.coverage.summary.tsv.bgz; associated downsampling data from https://storage.googleapis.com/gcp-public-data--gnomad/release/2.1.1/constraint/gnomad.v2.1.1.lof_metrics.downsamplings.txt.bgz; constraint metrics from Supplementary Dataset 11 from Karczewski et al. (Nature, 2020, https://static-content.springer.com/esm/art%3A10.1038%2Fs41586-020-2308-7/MediaObjects/41586_2020_2308_MOESM4_ESM.zip); cancer predisposition syndrome gene list - available in the suppl. for Rahman (Nature, 2014, https://www.ncbi.nlm.nih.gov/pmc/articles/PMC4975511/); and CPS mutations in adults from suppl. data 2 from Huang et al. (Cell, 2018, https://www.ncbi.nlm.nih.gov/pmc/articles/PMC5949147/). These links are also provided, when needed, in the comments for the code (see Code availability). Source data are provided with this paper.

## Code availability
The full, commented code[97] used for data wrangling, analysis, and visualization is available at (https://github.com/TruthSeqer/peds_cancer_genes_vs_constraint/blob/main/constraint_in_pCPS_genes_revised.Rmd). A readme file instructs users on how to fully reproduce all findings using the code and data. (https://github.com/TruthSeqer/peds_cancer_genes_vs_constraint/blob/main/README.md).

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

## Acknowledgements

We acknowledge the funders of this work; U.K.S. was supported by Rigshospitalets Forskningspuljer [grant: R186-A8440]; U.K.S., K.S., K.W., J.F.S., & T.H. were supported by Danish Cancer Society [grant: R-257-A14720] & Danish Childhood Cancer Foundation [grant: 2019-5934 & 2020-5769]; S.R. was supported by the Novo Nordisk Foundation [grant: NNF14CC0001]; K.J.K. was supported by the Novo Nordisk Foundation [grant: NNF21SA0072102]. Funders were in no way involved in the conduct of the study.

## Author contributions

Conceptualization: U.K.S. Data curation: U.K.S. Formal analysis: U.K.S. Funding acquisition: U.K.S., K.S., K.W. Investigation: All authors. Methodology: U.K.S., J.F.S. Project administration: U.K.S., T.V.O.H., K.S., K.W., K.J.K. Resources: U.K.S., T.V.O.H., K.S., K.W., K.J.K. Software: U.K.S. Supervision: T.V.O.H., S.R., K.W., K.S., K.J.K. Visualization: U.K.S. Writing - original draft: UKS, JFS. Writing - review and editing: All authors.

## Competing interests

K.J.K. is a consultant for Tome Biosciences and Vor Biosciences, and a member of the scientific advisory board of Nurture Genomics. Remaining authors have declared no competing interests.
