## [Peer Review File · Nature Communications]

REVIEWER COMMENTS

Reviewer #1 (Remarks to the Author): Expert in childhood cancer genomics and bioinformatics

The manuscript by Dr. Stoltze and colleagues provides an important contribution to the field of pediatric cancer predisposition and insightful findings on pathogenic germline variants in general. Based on the recently published mutational constraint spectrum, the authors elaborate on the evolutionary theory of genetic childhood cancer risk and apply these insights to review cancer predisposing genes and syndromes that deviate significantly from this theory. It is a solid piece of work presented in a clear, well-written paper, perhaps a bit too wordy in some of the Results sections. Apart from the presented data and analyses, I think the most important finding is that the authors confirm and emphasize the value of LoF constraint in prioritizing candidate genes for studying childhood cancer predisposition. Constraint has typically not been considered in the discovery of pediatric CPS genes, a practice the authors hope to change with their findings presented here.

Overall, the study is well conducted and it adds value to the current literature in going a few steps beyond a merely descriptive approach and reporting new evidence on known and novel pediatric cancer predisposition genes. However, some aspects of the manuscript should be improved or clarified:

1. Has the overlap in patients between these 11 pediatric pan-cancer studies been taken into consideration? Note that sample sets from Gröbner et al. (Nature2018) have substantial overlap with samples from Zhang et al. (NEJM2015), specifically for cancer samples collected from St. Jude. I cannot find any clarifications in the Methods and assume these might have been double-counted (including their germline mutations). In addition, please clarify: "Across 11 pediatric pan-cancer studies covering 4,833 children with cancer, ..." stated in the Results, but in the Abstract "4,810 children with cancer" are mentioned.

2. Ideally, a unified germline variant calling pipeline for point mutations and indels would have been applied to all samples, yet I do understand that this is very time- and resource consuming. However, simply taking germline call sets from the respective publication as they were reported in supplementary tables is probably too simplistic. At least re-annotation of all variants (after mapping to the same human genome build) with the latest version of VEP (Ensembl Variant Effect Predictor), for example, would be advised. Splice-sites are notoriously wrongly annotated, especially in older studies. Ideally, add also the latest version of ClinVar annotation regarding variant pathogenicity instead of relying on the relevant annotation from these (older) publications.

Some studies only report germline variants in the known-at-the-time CPS genes (and not yet for newer genes like ELP1 and GPR161, both reported only in 2020), some on all genes. Across the 11 pediatric pan-cancer studies, which germline variant call sets were reported (CPS only, all genes, etc.)?

Finally, from a methodological perspective (and referring to section “Methods” in the Supplement), I find the descriptions provided too short to fully understand and reproduce the analyses.

3. Regarding the repeated random sampling of 85 human genes, I would strongly suggest to sample only from the pool of protein-coding genes to better resemble pCPS genes, i.e., exclude pseudogenes, non-coding RNAs, and readthrough transcripts (like PMF1-BGLAP). For example, use only the "protein_coding" Gene/Transcript Biotype from GENCODE & Ensemble.

4. Please clearly define LOE ratio at the beginning of “Results” when used for the first time:

LoF observed vs. expected. It is currently only defined in the Figure legends, not the main text.

5. In Fig2, the last gene on the right side seems to be missing the dot and error bars in panels A-C (see RMRP, for example).

6. In Fig6, colors for missense and synonymous variants are hard to distinguish, might anyway be fine to not show synonymous variants in this plot.

7. Use either ELP1 (used in Fig5 and Fig8, for example) or IKBKAP (used in Figure2, etc.) in all Figures.

8. The sections “Reverting the effects of childhood cancer on the human gene pool” and “Gene discovery driven by evidence of evolutionary constraint” should be part of the Discussion/Conclusion, not the Results, since too speculative.

9. Replace “cooperated” with “corroborated” in this sentence: “Recently, this was cooperated in a unselected prospective study of WT which found that 5 out of 126 children with WT (4%) carry monoallelic germline LoF mutations in the gene.”

Reviewer #3 (Remarks to the Author): Expert in clinical genetics and genomics of childhood cancers, cancer predisposition, and prenatal screening

This paper uses existing datasets of variation databases and lists of curated cancer predisposition genes to evaluate whether loss of function variants in cancer predisposition genes are constrained in these population databases. There are some interesting findings in the paper which provide useful information, particularly insight into childhood penetrance of some of the more newly described genes. However, there are major concerns that the authors appear to be unaware of the substantial use of constraint for evaluating genes for other childhood onset disorders and thus many of the claims of primacy appear to be misplaced. Similarly, that genes with gain of function variants don't show constraint also seems obvious and could be a control for their model to describe early in the paper. Of importance, the gene list needs to be much more carefully curated by the authors as noted in comments below. Please note that the first two concerns relate to minor aspects of the paper but are critical to this reviewer given the author's apparent lack of appreciation of how these aspects of the paper would be understood very negatively by the readership and parents of children with cancer.

1. Most parents of children with cancer would argue strongly that this sentence is very insensitive to their loss. Childhood cancer has many tragic consequences of which any impact on the gene pool may appear to be minor and certainly not the most fundamental way that it differs from adult cancer. I strongly suggest rewriting this sentence to say that one unexplored aspect of the difference.

“While all of these features are in themselves enough to accentuate the unique biology of childhood cancer, the most fundamental way in which cancer in childhood differs from its adult counterparts is virtually unexplored”.

2. I would strongly recommend dropping the “Reverting the effects of childhood cancer on the human gene pool” section. It has several problems, perhaps most importantly a strong sense of eugenics that the field of genetics has suffered from often in the past (see recent ASHG report - <https://www.ashg.org/publications-news/press-releases/ashg-documents-and-apologizes-for-past-harms-of-human-genetics-research-commits-to-building-an-equitable-future/>) . In addition, other problems include (1) an overestimate of how successful we are in treating these cancers. Many patients with these CPS die of their first (if not their second) malignancy, (2) very large portions of the global population have no access to cancer treatment, let alone the reproductive technologies mentioned and (3) the severity of the treatment needed to cure these malignancies often directly reduces the fertility of patients.

3. The introduction seems to completely ignore the extensive work done in the last ten years on gene constraint in genes that result in many other severe pediatric onset Mendelian disorders, e.g. neurodevelopmental disorders. In fact, the gnomAD constraint scores (LOEUF) are so useful because they help to identify these genes. The authors imply that the concept that pediatric disorders that result in early death or lack of reproduction will show constraint in databases is a new idea. See for example this review from 2019 on new Mendelian disease discovery – Bamshad et al The American Journal of

Human Genetics 105, 448–455, September 5, 2019. Similarly, the figure of the WWII bombers is totally unnecessary given the familiarity of the field to this concept.

4. It is important in the writing to be clear when the authors are talking about a comparison with the average of all other genes versus comparisons with all other genes (see line 94-95). As noted above, there are other disease genes that show extreme constraints, particularly those that result in other severe childhood disorders that result in a lack of reproduction. They did random samplings of 85 genes. They did not do a comparison of 85 genes associated with other severe phenotypes.

5. The authors note that several of the genes show significant constraints in fact are responsible for both cancer and neurodevelopmental disorders, e.g, TSC1, TSC2, SMARCA4. They try to distinguish which genes are cancer only although this list is still highly problematic. For example, ALK is likely constrained due to the hypoventilation syndrome and the neuroblastoma syndrome is the result of missense variants – not LOF variants. Another example is CDH1 where children can have a complex phenotype - CLEFT LIP AND WITH OR WITHOUT CLEFT PALATE OMIM #137215 - but only develop cancers later in adulthood past most reproductive ages and thus the constraint is not clearly secondary to the cancer phenotype. The SMARC genes play an important role in Coffin-Siris syndrome. Although I have no doubt that genes which impact pediatric cancer alone can result in selection, e.g., RB1, this list of pediatric cancer only genes needs much more careful curation by individuals familiar with medical genetics.

6. Given that the authors emphasize the selection for SMARCA4, they may want to highlight the recent article demonstrating an increased risk of neuroblastoma and SMARCA4 – (Witkowski et al., J. Med Genet., 2023) a pediatric cancer association not previously noted, and which could potentially be responsible for some of the selection.

7. It is not clear how useful the missense variant analysis described here is to the paper. There is a need for regional missense constraint scores for this type of analysis to be more useful, particularly given the large size of many cancer susceptibility genes. The description is somewhat misleading given the nature of pathogenic variants as for TP53, VHL, RET a substantial portion of pathogenic alleles are missense alleles. However, these genes demonstrate cancer in both pediatric and adult age ranges and thus may not show constraint in childhood. The relationship between LOF and missense constraint should also be tempered – for example this sentence – “meaning that the AD/XLR pCPS genes that were (likely) not constrained for pLoF mutations also showed no difference in missense mutation constraint” - seems to ignore that RET is likely constrained for LOF variants due to Hirschsprung’s disease (a non-cancer condition) and the gain for function missense variants in RET cause the cancer phenotype.

8. Related to number 7, the point that some of the CPS genes, e.g. HRAS, are well documented to have gain of function variants as the disease mechanism. This is not a surprising finding and much more of a

confirmation of the validity of the analysis. This data should be raised much earlier in the paper and their lack of constraint described as essentially a control for the analysis not a new finding.

9. The description of the analysis for ELP1 and GPR161 is very interesting as these are more recently described genes and the field is trying to understand the appropriate surveillance for these findings. Isn't the most likely reason for the lack of constraint for both genes low penetrance. Why is positive selective pressure described for ELP1 and not GPR161? Please clarify this analysis.

10. The description of the association of MSH2 with constraint is very interesting and as noted is consistent with the recent meta-analysis by Kratz et al., JNCI, 2022. However, it is likely that the accumulation of pediatric cancer studies used in the analysis here strongly overlaps with the studies put together in the Kraft et al paper to complete their analysis. The authors should carefully review these two cohorts to confirm that there is no redundant data. This is particularly important as several recent studies of Lynch syndrome diagnoses (not CMMRD) in pediatric cancer (not cited by the authors) did not show any prevalence for MSH2 as the causative gene and patients are described with all of the MMR genes – Fiala EM et al., Nat Cancer, 2022, Scollon, S et al., Ped Blood Cancer, 2022, and Macarthur, TA et al., J Pediatr Surg Case Rep, 2022.

11. The historical axes figure and discussion seems unnecessary. Discovery is often a function of methodologies available at the time. The earlier discoveries often required large families which one would argue are less likely to be constrained genes. Exome sequencing of trios looking for de novo variants (likely to be constrained genes) developed later. Penetrance is another important aspect of gene/disease discovery and would favor more constrained genes. Certainly, all discovery pipelines now uniformly look carefully at constrained genes for severe pediatric phenotypes (whether cancer or others) so this again seems like an unnecessary section to an already long paper.

Minor:

1. Figure 3 needs appropriate legends (what do the colors stand for) and definition of the categories for the reader to follow. What is the X axis designating? Similarly, Figure 6 A is missing the genes names on the X axis.

Reviewer #4 (Remarks to the Author): Expert in cancer genetics and evolution, and population genetics

Stoltze et al. aimed to quantify the selective constraint on genes associated with pediatric cancer predisposition syndrome (pCPS) in their paper. Given that most pediatric cancers are linked to poor survival rates, which nearly preclude reproduction, genes with mutations that increase cancer risk during childhood are expected to be under strong selective constraint. By focusing on the loss-of-function observed/expected upper bound fraction (LOEUF) metric reported in the gnomAD database, the authors indeed found a severe depletion of unique loss-of-function (LoF) variants in pCPS genes observed in adult humans than neutral expectation. For some of these pCPS genes, they also observed seemingly enrichment of LoF mutations in children with cancer. The authors then discussed a few autosomal dominant pCPS genes that lack evidence of constraint, as well as two autosomal recessive pCPS genes that show strong evidence of constraint, which suggest false association to childhood cancer risk or incorrect mode of inheritance.

It is a nice idea to examine the selection constraint of pCPS genes, and I am largely convinced these genes are subjected to quite strong selective pressure. However, it is worth noting that the LOEUF metric only measures selective constraint on LoF variants and has several limitations, including incomplete and biased power across genes. The biased power suggests that it is crucial to match different sets of genes to avoid potential confounders, while the incomplete power means that although genes with low LOEUF scores are highly constrained, but not all constrained genes necessarily have low LOEUF scores. As a result, it requires caution to conclude that certain gene lacks selective constraint. In addition, the method that the authors used to quantify enrichment of mutations in children with cancer is highly problematic, so I am skeptical about findings from this analysis. Lastly, the methods are not sufficiently described in a couple of places, and the figures could be improved. My specific comments are outlined below.

Major points:

1. Limitations of using LOEUF score for measuring selective constraint

First, this metric is purely based on loss-of-function variants, so it only detects signal when complete loss of one copy of the gene (or 50% reduction in gene expression) is deleterious, but this metric cannot measure selective constraint on gain-of-function mutations or other types of mutations. This is a particular concern in the analysis of cancer driver genes, because cancer driver genes include both tumor suppressor genes, the loss of which leads to cancer, and oncogenes, the activation of which leads or predisposes to cancer. Can the author break the pCPS genes into two classes, tumor suppressor genes and oncogenes, and analyze them separately? My expectation is that tumor suppressor pCPS genes would be more enriched for LOEUF-based constrained genes compared to random gene sets, although the oncogenes may not be (this does not mean that the latter group is not constrained; it just means that these genes are not significantly depleted for LoF variants). Along this line, this expectation fits the observation for SAMD9 and SAMD9L, pathogenic variants in which are gain-of-function. I hope the authors could discuss the distinction between oncogene and tumor suppressor genes earlier.

Second, LOEUF has different powerful in detecting signal of selective constraint from gene to gene. For example, one specific bias is much reduced power for short genes, because it is challenging to detect a significant depletion from expectation when the expectation is already low. Comparing Fig2 A and B, it seems pCPS genes tend to be associated with narrower confidence intervals compared to the random set, possibly because pCPS genes are enriched for long genes. In addition to gene length, other genomic features of a gene may also bias the LOEUF's detection power. Given this bias, it would be nice if the authors can match for gene length and other features when drawing random genes to compare with pCPS (for example, analysis corresponding to Fig 2AD). Similar concerns apply to the O/E ratio of missense mutations shown in Fig 7, the comparisons between high vs low risk pCPS genes and between high vs. low mutation rate pCPS genes, and comparison between adult and childhood pCPS genes.

Lastly, LOEUF relies solely on number of unique LoF variants observed in the same but ignores the allele frequencies of the observed variants, so it has little to no power in detecting selective constraints on variants with recessive fitness effects, the signal of which is lower allele frequency than neutral expectation. Accordingly, the statement in lines 102-104 is inaccurate: technically, genes where LoF variants have completely recessive effects are also constrained, which is manifested in lower frequencies of deleterious mutations, although this depletion is likely not picked up by LOEUF.

2. Estimating number of LoF variants in children with cancer by extrapolation

The authors do not explicitly explain how the extrapolation was done, but based on the numbers provided in Fig 5, it seems they assumed that the number of LoF variants observed would be proportional to the sample size (i.e., the number of children) and compared the scaled number to the expectation and observation in adults. This extrapolation method is incorrect, because both the expectation and observation (in adult human) reported by gnomAD are the number of unique LoF variants, which does NOT scale with the sample size. In fact, this number is expected to increase slower as the sample size grows larger, which can lead to the apparent "enrichment" of LoF mutations in children with cancer when the number is simply scaled up. Instead of extrapolation, the author can look at dataset with WGS data available (such as UK10K), down-sample the adult data randomly (multiple times) to match the number of children, and ask how often the number of unique LoF observed in adult samples is lower than observed in pediatric cancer patients. Alternatively, the authors can compare the average number of LoF per individual between adults in gnomAD and children with cancer, by summing the allele frequencies of all LoF variants observed in that group. This approach is unbiased with regard to the sample size, but the authors need to account for uncertainty properly especially for genes with small sample size. If the results still hold, this is evidence for "evolution in action", but it is still an over-statement that the increased cancer risk is what is driving natural selection (lines 155-157).

3. Selection constraint driven by childhood cancer risk

The authors found that pCPS genes with neoplasm as the only known phenotype are still highly constrained and concluded that “stand-alone early life cancer risk can drive natural selection” (lines 139-140; similarly in lines 185-186). However, this conclusion is premature, because mutations in these genes can still be pleiotropic, causing higher cancer risk and other weak phenotypes simultaneously, and the subtle, subclinical phenotypes may lead to severe reduction in fitness (for detailed discussion of phenotypic vs. fitness effect, please see Fuller et al., 2019 Nat Genet). In fact, most constrained genes are not associated with disease or fertility problem, suggesting that subtle, clinical phenotypes can contribute to fitness loss (Gardner et al 2022 Nature).

4. Discussion of DIS3L2 and MSH2

In the abstract, the authors wrote that monoallelic LoF mutations in these two genes may increase childhood cancer risk, but this conclusion was not explicitly stated in the main text of the corresponding section. The presence of selective constraint alone is supporting evidence but not sufficient, as these genes may be truly recessive with regard to cancer risk, but heterozygotes carrying one defective gene copy may have other weak, subclinical phenotypes that reduce fitness subtly, leading to selective constraint (for example, even 5% reduce in fertility is considered evolutionary highly deleterious). A more compelling piece of evidence is enrichment of heterozygotes in children with cancer, but why does PMS2 lack selective constraint despite being enriched in children with cancer? Also, how do the authors reconcile the observations that heterozygous DIS3L2 LoF mutations were seen in 5 out of 126 children with Wilms tumor but none observed in the 3045 patients across 11 pediatric pan-cancer studies? How to interpret the observation of only one overlapping LoF mutation between children with WT and adults? The authors need to state their argument, explanation, and conclusion more clearly in this section.

5. Questions regarding methods

1) Fig 1: For the definition of “Likely constrained” and “Likely not constrained” genes, is there additional criteria in addition to $LOE < 0.35$ vs $LOE > 0.35$? If not, is it meaningful to classify genes into these two categories purely based on the LOE value, for genes with large confidence intervals?

2) In the simulation of 1000-year-survival change of mutations in pCPS genes, is it assumed that the cancer phenotype, once manifested, is incompatible with survival/reproduction? Is this true for all or most types of childhood cancer?

3) How is the weighted ratio on the y-axis calculated in Fig10A? How many unique genes are involved for the 229 and 485 pathogenic mutations found in adults and children with cancer? Why is it meaningful to divide the genes in four groups, when the distinction between the two middle groups is somewhat stochastic?

4) How do the authors translate the 1,047 cancer risk SNPs into 722 genes, especially for SNPs not found in gene body?

Minor comments:

In lines 50-53, the five bullet points should be all verbs so that are parallel to each other.

Fig6: it is not obvious to me what this figure is meant to show and why it is cited in the section “pCPS genes with monoallelic cancer risk as the sole phenotype show constraint”.

Fig 9: Is the order of genes in Fig 9A meaningful? If not, can the authors sort the genes by their LOEUF score to avoid crossing of the arrow heads? What does TUM (corresponding to triangles) stand for? What is the interpretation of results shown in Fig 9B?

For a manuscript of this length, there are too many figures. I would suggest the authors remove or combine some of them (5-6 figures would be ideal).

Line 359: the word “both” should be removed, as three types of biases are discussed.

Line 388: “form” should be “from”.

Reviewer #5 (Remarks to the Author): Expert in bioethics of reproductive technologies, prenatal selection and screening, and genomic medicine

This paper focuses on the population genetics of childhood cancer, and makes some remarks about possible interventions and their implications for the human gene pool. Briefly, the authors suggest that the link between genes associated with childhood cancer and ‘constraint’ (an unexpectedly low rate of mutation) is explained by natural selection. Developing treatments that cure the phenotype of childhood cancer, thereby allowing children carrying those genes to live to sexual maturity and pass their cancer-causing genes on, will act against natural selective pressure, resulting in those mutated genes eventually

becoming more common in the population. To counterbalance this effect they suggest the use of a variety of prenatal screening methods (population based screening, preimplantation genetic diagnosis of in vitro-produced embryos, and “other family planning interventions”) to ensure those future individuals still carrying the mutated genes do not appear in the population. They argue that “Medically deselecting embryos with variants in highly constrained genes effectively mirrors the selection process of nature evident in the human gene pool today.”

In other words, those children who are already born will be treated to enable them to survive their cancer, or other genetically related disease; but we will act (or at least, offer people the opportunity to act) to prevent further individuals from being born and going on to develop a (presumably now treatable) cancer.

It is incontrovertible that curing paediatric cancer is a good thing. There is more ambiguity over the moral rightness of intervening prenatally to ensure that people with particular traits, like paediatric cancer, are not born. Bioethicists specialising in disability have developed a critique of prenatal selection against disabling conditions, arguing that at times it can be nothing more than discrimination against people with disabilities that are perfectly compatible with a good and flourishing life. But on the whole they aren't making this argument about conditions as severe and potentially lethal as paediatric cancer.

Nevertheless, there are still some major ethical considerations to be discussed. For one thing, prenatal identification and selection, by whatever method, effectively becomes a built-in part of the management of certain paediatric cancers (for the survivors once they are adult); this may be the morally right thing to do for the individual, but has implications for the routinisation of selection for the community as a whole.

Second, in contemporary biomedicine there are many situations in which a proxy – such as a gene locus – becomes a marker for the thing itself. The fact that the marker is some steps away from the phenomenon it stands for can easily slip out of sight. In prenatal genetic testing and screening, an allele stands for a predicted experience of suffering or disadvantage that we consider severe enough to call ‘disability’ and to warrant termination of pregnancy. For obvious reasons, in this situation use of the gene locus as a proxy is unavoidable as it is the only information we have; but the extent to which it's an accurate predictor of future suffering or disadvantage depends on numerous factors, including social and environmental ones. The paper suggests that the logic of observing that genes causing paediatric cancer are ‘constrained’ could be reversed: that if genes are found to be constrained then that might be an indicator that their mutated versions are likely to be pathological. As the authors paper note, there are around 3000 human genes that show constraint but the majority of these are not (yet) linked to pathology. But it is not inconceivable that a gene showing mutational constraint will come to be considered suspect in itself even if no link with disease has yet been shown, and that its presence will be assumed to raise the risk profile of the carrier. There should be some caution about a growing reliance

on markers that are more and more distanced from experiencing the pathology they are supposed to indicate.

A final point to think about: the rationale the paper gives for using prenatal selection in these cases is essentially about avoiding long-term, unwanted changes in the human gene pool. In stark contrast, most population prenatal screening today is offered on the basis of the lives of individuals: enhancing individual reproductive autonomy and avoiding suffering (of the future child). Screening programs strenuously avoid the population-level arguments about the quality of the gene pool reminiscent of the historical thinking and practices that today would be condemned as overtly eugenic. Whether or not prenatal reproductive selection can justifiably be called 'eugenic', as some ethicists and activists do, is a highly contested point and not one I'm going to address here. But it is important to remain alert to the risks, both actual and perceived, of a scientific advance that fundamentally shifts the moral framework within which people consider reproductive choice.

To the reviewers,

We are immensely grateful for the opportunity to address the comments and input of the four expert reviewers whose important comments both clarified and strengthened our manuscript.

As the reviewer comments were extensive, we have provided a summary here listing the main changes made to the revised manuscript, including its methods section. This is in addition to the detailed point-by-point response in the following pages:

1. **Random sampling of genes.** In order to increase the validity of the random samples of genes, we i) compared the gene length of the 85 genes related to childhood cancer to those of all genes, and, in light of a clear disparity, we implemented a random sampling that matches the sets based on gene size. Method and Results text, as well as associated figures, have been updated accordingly. We consider this change a technical clarification, and importantly, the conclusions drawn remain unaltered from the original submission.
2. **Differentiation between AD/XLR pCPSs driven by gain-of-function vs. loss-function variants.** For a total of nine genes, all with autosomal dominant mode of inheritance, the genotypes driving the pCPS was GoF. These genes are treated as an independent group and all text, tests and figures have been updated accordingly.
3. **Expanded quality control regarding pathogenic variants in pediatric pancancer studies.** We accounted for a 259-patient overlap between two major pediatric pancancer studies (Zhang, NEJM2015 & Grobner, Nature2018). Method and Results text, as well as associated figures, have been updated accordingly. This change provided an important improvement in accuracy, yet it was ultimately without any impact on the conclusions drawn.
4. **Changes regarding ethically sensitive subjects.** Based on your suggestions we removed sections regarding the iatrogenic effects on genetic constraint of pCPS genes and reworded the introduction, where relevant.
5. **Other figure and text reduction.** In addition to the cuts made above, we have shortened the text and downsized/rearranged figures so that figure 3 & 4 and 6 & 7 from the original submission is now each one figure (Figure 3 and 5, respectively) in the revision. This also included a restructuring to divide the main text into results and discussion as per editorial guidelines.

Both these and all other changes/comments are addressed in full detail below. Collectively, we are convinced the implemented changes have greatly improved the clarity and technical aspects of the manuscript, yet, the original conclusions remain intact and, arguably, stand stronger following the numerous helpful suggestions provided by the four reviewers.

With gratitude and sincere wishes for a good summer (or winter for those below the equator),

Corresponding Author Dr. Ulrik Stoltze & all Co-authors

Reviewer #1 (Remarks to the Author): Expert in childhood cancer genomics and bioinformatics

The manuscript by Dr. Stoltze and colleagues provides an important contribution to the field of pediatric cancer predisposition and insightful findings on pathogenic germline variants in general. Based on the recently published mutational constraint spectrum, the authors elaborate on the evolutionary theory of genetic childhood cancer risk and apply these insights to review cancer predisposing genes and syndromes that deviate significantly from this theory. It is a solid piece of work presented in a clear, well-written paper, perhaps a bit too wordy in some of the Results sections. Apart from the presented data and analyses, I think the most important finding is that the authors confirm and emphasize the value of LoF constraint in prioritizing candidate genes for studying childhood cancer predisposition. Constraint has typically not been considered in the discovery of pediatric CPS genes, a practice the authors hope to change with their findings presented here.

Overall, the study is well conducted and it adds value to the current literature in going a few steps beyond a merely descriptive approach and reporting new evidence on known and novel pediatric cancer predisposition genes. However, some aspects of the manuscript should be improved or clarified:

Thank you for this very accurate recap of our manuscript and its main findings. We greatly appreciate the time you have taken to thoroughly review our work and that you consider it an important contribution to the field of pediatric cancer predisposition.

1. Has the overlap in patients between these 11 pediatric pan-cancer studies been taken into consideration? Note that sample sets from Gröbner et al. (Nature2018) have substantial overlap with samples from Zhang et al. (NEJM2015), specifically for cancer samples collected from St. Jude. I cannot find any clarifications in the Methods and assume these might have been double-counted (including their germline mutations). In addition, please clarify: "Across 11 pediatric pan-cancer studies covering 4,833 children with cancer, ..." stated in the Results, but in the Abstract "4,810 children with cancer" are mentioned.

Thank you for pointing us to this important oversight. No, the overlap has indeed not been addressed, and we agree that this introduces a double-reporting bias. Gröbner et al. (Nature2018) note that "[out of] n = 914 individual patients, about 25% of samples overlapping with the previous study". Although not explicitly stated, we have assumed that this refers to the 259 samples [28.4%, 259/914] that have a St. Jude ID (SJxxxxnnn). Following your input, we have 1) removed variants reported by Zhang et al. (NEJM2015) for this subset of patients and 2) subtracted 259 from the total n of patients in Zhang et al. (NEJM2015) — effectually only keeping the Gröbner et al. (Nature2018) data for the 259 samples in question, as the more recent study was considered more likely to be of higher accuracy. Of note, this did run the risk of excluding relevant variants (i.e., variants in the 85 pCPS

genes), because Zhang et al. (NEJM2015) used a far more extensive germline gene panel, which did include five pCPS genes not on the panel used by Grobner et al. (Nature2018). However, there were no reported variants in these gene in the 259 samples of interest. In the end, the updated pipeline resulted in changes to the total number of patients and the total number of pathogenic variants, now 4,574 and 267, respectively (see revised figures 5 & 8). We have expanded the method section to reflect these updates and point to the relevant section in the accompanying code. The updated numbers have been corrected throughout the revised manuscript and updated in the relevant figures. The correction also mitigated the inconsistency you correctly pointed out in your comment.

2. Ideally, a unified germline variant calling pipeline for point mutations and indels would have been applied to all samples, yet I do understand that this is very time- and resource consuming. However, simply taking germline call sets from the respective publication as they were reported in supplementary tables is probably too simplistic. At least re-annotation of all variants (after mapping to the same human genome build) with the latest version of VEP (Ensembl Variant Effect Predictor), for example, would be advised. Splice-sites are notoriously wrongly annotated, especially in older studies. Ideally, add also the latest version of ClinVar annotation regarding variant pathogenicity instead of relying on the relevant annotation from these (older) publications.

Some studies only report germline variants in the known-at-the-time CPS genes (and not yet for newer genes like ELP1 and GPR161, both reported only in 2020), some on all genes. Across the 11 pediatric pan-cancer studies, which germline variant call sets were reported (CPS only, all genes, etc.)?

Finally, from a methodological perspective (and referring to section “Methods” in the Supplement), I find the descriptions provided too short to fully understand and reproduce the analyses.

We appreciate that you raise this highly valid point for discussion. We fully agree that the pathogenicities of variant classifications reported in pediatric pancancer studies would, for a subset, likely be reclassified using more recent standards. However, there are several reasons why such a reclassification is outside the scope of the current study. Most importantly, the primary reason we compile data across the studies was to quantify LoF variants (i.e., protein truncating variants), regardless of their pathogenicity (although such variants of course often are pathogenic). Only as a secondary result, we found it instructive to the reader to illustrate the variants from the literature - as reported - in a figure. Precisely due to classification bias, not to mention reporting bias/differences, we do not use pathogenicity to make any conclusions in our study. We have made changes and expansions to the manuscript text to clarify this important point, so that variants are referred to only as ‘reported as pathogenic’.

Re “Some studies only report germline variants in the known-at-the-time CPS genes”: This is true, and we carefully considered this with comparisons of each gene panel used on a per-study basis (see Supplementary Data 16-26, as well as the final column in figure 5 (original submission) with the numbers reflecting only studies that included the relevant gene on their panel.

Re “Across the 11 pediatric pan-cancer studies, which germline variant call sets were reported”: Only the variants on the germline panel employed, see Supplementary Data 5-15 that detail this for each study. We recognize that some variants may not have been reported, i.e., reporting bias. However,

this would only lead to an underestimation of variation observed in childhood cancer patients, which would lessen the observed difference from the human pangenome.

We agree that these and other related methodical aspects were not sufficiently described in the original submission, and hence, we have greatly expanded details, which should be reproducible based on the data in Supplementary Data 5-26 in combination with the accompanying code.

3. Regarding the repeated random sampling of 85 human genes, I would strongly suggest to sample only from the pool of protein-coding genes to better resemble pCPS genes, i.e., exclude pseudogenes, non-coding RNAs, and readthrough transcripts (like PMF1-BGLAP). For example, use only the "protein_coding" Gene/Transcript Biotype from GENCODE & Ensemble.

Thank you. We agree with this important point, and this is already the case; the original random sampling was of the genes for which constraint metrics were calculated by Karczewski et al (Nature, 2020), in which the transcripts were all annotated as ‘protein_coding’. Yet, other reviewers have pointed to gene sizes as being an essential metric to control for. To address this concern, we have added analyses with comparisons to a size-matched subset of genes and this update impacts your general point, i.e., making the random samples more representative of the known pCPS genes.

4. Please clearly define LOE ratio at the beginning of “Results” when used for the first time: LoF observed vs. expected. It is currently only defined in the Figure legends, not the main text.

Thank you for highlighting this source of potential confusion; in the revised manuscript, the LOE ratio is now clearly defined when used the first time in the Results section.

5. In Fig2, the last gene on the right side seems to be missing the dot and error bars in panels A-C (see RMRP, for example).

This is correct; the *RMRP* gene is a non-coding RNA gene defined as category 1 in terms of causing a pediatric CPS. As such, it is, for the sake of completion included in our work, but of note it was not included in Karczewski et al., Nature, 2020, as it is not ‘protein_coding’ (cf. your point #3 above). To clarify this, we have now added an explaining remark in the figure legend and the text in the supplementary materials.

6. In Fig6, colors for missense and synonymous variants are hard to distinguish, might anyway be fine to not show synonymous variants in this plot.

We very much agree with your comment, the colors are not distinguishable. After review of the total numbers of reported pathogenic variants (see response to your points 1 & 2 above), we have recrafted the figure using the updated numbers. Following your suggestion, we also elected to exclude pathogenic synonymous variants (this involved only one variant, and these may in fact be functionally splice-altering/protein-truncating). This was clarified in the figure caption.

7. Use either *ELP1* (used in Fig5 and Fig8, for example) or *IKBKAP* (used in Figure2, etc.) in all Figures.

We agree. *IKBKAP* is the name used in the data from Karczewski et al., Nature, 2020, yet based on your suggestion, we have elected to use *ELP1* throughout the revised manuscript, as this is the gene name known in the field of pediatric oncology and used by Waszak et al., Nature, 2020 and also the canonical name used by HGNC.

8. The sections “Reverting the effects of childhood cancer on the human gene pool” and “Gene discovery driven by evidence of evolutionary constraint” should be part of the Discussion/Conclusion, not the Results, since too speculative.

Yes, this is astute, and we agree that ‘Results’ was not an appropriate heading for these two sections. This issue has been resolved, cf. points 4 and 5 at the very beginning of this document.

9. Replace “cooperated” with “corroborated” in this sentence: “Recently, this was cooperated in a unselected prospective study of WT which found that 5 out of 126 children with WT (4%) carry monoallelic germline LoF mutations in the gene.”

This has been updated according to your suggestion in the revised manuscript.

Reviewer #3 (Remarks to the Author): Expert in clinical genetics and genomics of childhood cancers, cancer predisposition, and prenatal screening

This paper uses existing datasets of variation databases and lists of curated cancer predisposition genes to evaluate whether loss of function variants in cancer predisposition genes are constrained in these population databases. There are some interesting findings in the paper which provide useful information, particularly insight into childhood penetrance of some of the more newly described genes. However, there are major concerns that the authors appear to be unaware of the substantial use of constraint for evaluating genes for other childhood onset disorders and thus many of the claims of primacy appear to be misplaced. Similarly, that genes with gain of function variants don't show constraint also seems obvious and could be a control for their model to describe early in the paper. Of importance, the gene list needs to be much more carefully curated by the authors as noted in comments below. Please note that the first two concerns relate to minor aspects of the paper but are critical to this reviewer given the author's apparent lack of appreciation of how these aspects of the paper would be understood very negatively by the readership and parents of children with cancer.

Thank you for taking the time to thoroughly review our work, for your accurate recap of our manuscript and, importantly, for raising these valid discussion points. In the following, we provide discussion of and response to your comments in a point-by-point fashion. We are truly grateful for the many insights and improvements they resulted in.

1. Most parents of children with cancer would argue strongly that this sentence is very insensitive to their loss. Childhood cancer has many tragic consequences of which any impact on the gene pool may appear to be minor and certainly not the most fundamental way that it differs from adult cancer. I strongly suggest rewriting this sentence to say that one unexplored aspect of the difference. “While all of these features are in themselves enough to accentuate the unique biology of childhood cancer, the most fundamental way in which cancer in childhood differs from its adult counterparts is virtually unexplored”.

Thank you for alerting us to this reading of this introductory sentence. Being mindful of this, we have entirely reworded the section: “A sixth feature is that pediatric cancers, by definition, occur during childhood. Tragically, such cancers tend, especially historically, to cause death prior to reproduction (ref 27). Consequently, the transmission of associated childhood cancer risk variants is, in theory, put under massive evolutionary pressure — an aspect that remains virtually unexplored.”

2. I would strongly recommend dropping the “Reverting the effects of childhood cancer on the human gene pool” section. It has several problems, perhaps most importantly a strong sense of eugenics that the field of genetics has suffered from often in the past (see recent ASHG report - <https://www.ashg.org/publications-news/press-releases/ashg-documents-and-apologizes-for-past-harms-of-human-genetics-research-commits-to-building-an-equitable-future/>) . In addition, other problems include (1) an overestimate of how successful we are in treating these cancers. Many patients with these CPS die of their first (if not their second) malignancy, (2) very large portions of the global population have no access to cancer treatment, let alone the reproductive technologies mentioned and (3) the severity of the treatment needed to cure these malignancies often directly reduces the fertility of patients.

Thank you for raising this important discussion. Along with the editors of the journal, we considered your comment carefully and, as you suggest, we decided to entirely remove the section and the associated illustration.

3. The introduction seems to completely ignore the extensive work done in the last ten years on gene constraint in genes that result in many other severe pediatric onset Mendelian disorders, e.g. neurodevelopmental disorders. In fact, the gnomAD constraint scores (LOEUF) are so useful because they help to identify these genes. The authors imply that the concept that pediatric disorders that result in early death or lack of reproduction will show constraint in databases is a new idea. See for example this review from 2019 on new Mendelian disease discovery – Bamshad et al The American Journal of Human Genetics 105, 448–455, September 5, 2019. Similarly, the figure of the WWII bombers is totally unnecessary given the familiarity of the field to this concept.

This is a very valid point with which we very much agree. Since the pre-printing and presentation of our work at scientific conferences, we have indeed become more aware of the prior work in other tragic pediatric phenotypes (beyond what is already highlighted in Karczewski et al., Nature, 2020). In accordance with your comment, we have updated the revised manuscript to include a dedicated paragraph in the introduction section outlining several examples, including autism, stillbirth, and rare diseases.

Still, based on the existing literature and our experiences from presenting and discussing our work with the scientific community, it is our clear impression that the concept of evolutionary constraint is novel to the field of genetic predisposition in pediatric oncology, for which it - as opposed to the other disorders now discussed in the revised manuscript - has remained unexplored until now.

Our interest in this field was initially sparked by our desire to explore the use of mutational constraint in the analysis of germline data from children with cancer. However, during our extensive literature search, we did not come across any scientific references discussing or qualifying the relevance of this approach to the phenotype in question. Therefore, it appears that the concept of mutational constraint, along with its implications for understanding and analyzing genetic pediatric cancer risk, is novel and not widely familiar to researchers and clinicians in the field of oncology, who are among the core audience for this paper.

While experts in human genetics and clinicians involved in the care of certain non-malignant childhood disorders may be acquainted with mutational constraint and its role, we believe that drawing parallels to survival bias may be beneficial for other readers who may not be intimately familiar with large-scale germline genomics, disease-gene discovery, and evolution.

We express concern that assuming a higher level of general familiarity with the subject of human mutational constraint and its relevance to genetic predisposition might compromise the clarity of our study, a quality that has been emphasized by other expert reviewers. Since we do not consider the figure essential to the core findings and conclusions of our work, we leave the final decision on its removal to the journal's Editors.

4. It is important in the writing to be clear when the authors are talking about a comparison with the average of all other genes versus comparisons with all other genes (see line 94-95). As noted above, there are other disease genes that show extreme constraints, particularly those that result in other severe childhood disorders that result in a lack of reproduction. They did random samplings of 85 genes. They did not do a comparison of 85 genes associated with other severe phenotypes.

Yes, this is correct, the random sampling was of the canonical transcripts of all genes included by Karczewski et al., Nature, 2020 (these were all protein coding). In accordance with your input, we have altered/added wording to make this clear in the revised manuscript. The aim of this study was to investigate if genes linked to childhood cancer risk are more constrained than other genes. Because

the aim was not to investigate whether genes linked to childhood cancer were more or less constrained than genes linked to other tragic phenotypes, such analyses were not considered within scope.

5. The authors note that several of the genes show significant constraints in fact are responsible for both cancer and neurodevelopmental disorders, e.g, TSC1, TSC2, SMARCA4. They try to distinguish which genes are cancer only although this list is still highly problematic. For example, ALK is likely constrained due to the hypoventilation syndrome and the neuroblastoma syndrome is the result of missense variants – not LOF variants. Another example is CDH1 where children can have a complex phenotype - CLEFT LIP AND WITH OR WITHOUT CLEFT PALATE OMIM #137215 - but only develop cancers later in adulthood past most reproductive ages and thus the constraint is not clearly secondary to the cancer phenotype. The SMARC genes play an important role in Coffin-Siris syndrome. Although I have no doubt that genes which impact pediatric cancer alone can result in selection, e.g., RB1, this list of pediatric cancer only genes needs much more careful curation by individuals familiar with medical genetics.

Thank you for this comment. We fully agree that the distinction between pCPS genes related to a pure cancer phenotype and pCPS genes with additional non-cancer phenotypes is crucial to this work and deserves closer scrutiny. We have therefore focalized this topic in our manuscript, with a level of attention we considered suitable. Below, we provide a comprehensive response to each part of your insightful comments, which in the end led to several changes — i.e., genes associated with CPS phenotypes GoF as well as the *CDH1* gene has been excluded from the analysis in question.

At the start of your comment, three examples of genes with pleiotropy beyond cancer susceptibility are listed, incl. *SMARC4A*. In our manuscript, *SMARC4A* was conversely designated as having an pure cancer risk phenotype. This designation is based on the observation that loss-of-function variants drive the phenotype Rhabdoid tumor predisposition syndrome 2 (MIM #613325). The gene is also associated with the phenotype Coffin-Siris syndrome 4 (MIM #614609), but contrary to the former this results from missense mutations eliciting gain-of-function (see GeneReviews book NBK131811). These are the only phenotypes listed for *SMARC4A* in OMIM, and as such, we considered our designation relevant, i.e., based on the published literature, the most credible driver of the mutational constraint of loss-of-function variants in *SMARC4A* is Rhabdoid tumor predisposition syndrome 2, which has no substantiated phenotype other than cancer. We have clarified this point for *SMARC4A* and other genes in the revision.

Next, *ALK* is highlighted, including its link to the hypoventilation syndrome and missense variants. We agree that, as the neuroblastoma susceptibility phenotype is driven only by gain-of-function genotypes, this gene should have been (and now is) separated out in our analysis, cf. point #8 of your comments. However, we also wish to point out that a hypoventilation phenotype is not substantiated for this gene (PMID: 36140661) as opposed to *PHOX2B*, which, while also causing neuroblastoma susceptibility (MIM#613013), is related to a well-known hypoventilation phenotype (MIM# 209880). Indeed, *PHOX2B* was not included among the original 23 genes with an isolated cancer

phenotype, precisely due to this as well as its associated Hirschsprung disease non-cancer phenotypes.

Next, you highlight *CDHI*. It is correct that the relevant phenotype is named “Diffuse gastric and lobular breast cancer syndrome with or without cleft lip and/or palate”. Importantly, the link to cleft palate is originally based on only two cases reported in 2006 (PMID: 15831593). Since then, this observation has only been minimally substantiated despite *CDHI* being the subject of immense scrutiny and widespread clinical testing due to the possibility of risk-reductive surgical interventions. Documented families in the literature count in the hundreds (PMID: 34952833) of which only a total of eight have been reported as having any cleft lip/palate phenotype (PMID: 30306390). While cleft lip/palate is perhaps somewhat enriched among germline *CDHI* carriers, in our view, this does not, overall, paint a picture of a “complex phenotype”. Of note, the penetrance for cancer before or in the reproductive years is 30-40% (PMID: 26182300). Thus, we, including clinical geneticists, respectfully do not agree that individuals with pathogenic *CDHI* mutations “only develop cancers later in adulthood past most reproductive ages”; as the associated cancers occur starting from the teenage years, with a median onset as low as 38 years (see GeneReviews book NBK1139), and hence risk-reductive surgical intervention is recommended to be considered already from 20 years of age. Ultimately, given the small number of cases reported with CLP, we, guided by your comment, elected to change the designation from ‘no non-cancer phenotype’ to ‘discrete non-cancer phenotype’, thus excluding it from the isolated cancer risk analysis.

Lastly, you mention that the SMARCA genes are related to Coffin-Siris syndrome; here, the argument for including *SMARCE1* and *SMARCB1* is the same as the one mentioned for *SMARCA4* above with reference to the GeneReviews book NBK131811 “Evidence indicates that pathogenic variants in *SMARCA4*, *SMARCB1*, and *SMARCE1* act through a gain-of-function mechanism, suggesting that large pathogenic deletions or duplications are unlikely to occur [...]”

Please allow us to reiterate that we are grateful for this discussion, which aided in clarifying the designations provided in Supplementary Data 3. We are confident that with the changes now implemented based on your highly relevant comment, the analysis of pCPS genes associated with isolated cancer risk, as a whole, supports the conclusion drawn.

6. Given that the authors emphasize the selection for *SMARCA4*, they may want to highlight the recent article demonstrating an increased risk of neuroblastoma and *SMARCA4* – (Witkowski et al., J. Med Genet., 2023) a pediatric cancer association not previously noted, and which could potentially be responsible for some of the selection.

Most certainly. Thank you for bringing this very relevant paper to our attention; we have added a reference to the study in the relevant section of the revised manuscript.

7. *It is not clear how useful the missense variant analysis described here is to the paper. There is a need for regional missense constraint scores for this type of analysis to be more useful, particularly*

given the large size of many cancer susceptibility genes. The description is somewhat misleading given the nature of pathogenic variants as for TP53, VHL, RET a substantial portion of pathogenic alleles are missense alleles. However, these genes demonstrate cancer in both pediatric and adult age ranges and thus may not show constraint in childhood. The relationship between LOF and missense constraint should also be tempered – for example this sentence – “meaning that the AD/XLR pCPS genes that were (likely) not constrained for pLoF mutations also showed no difference in missense mutation constraint” - seems to ignore that RET is likely constrained for LOF variants due to Hirschsprung’s disease (a non-cancer condition) and the gain for function missense variants in RET cause the cancer phenotype.

We very much agree that pangenomic analysis of missense variant spectrum presents a challenge compared to the LoF variants. This is why we begun that section with a paragraph which highlights those points;

“Yet, the sheer number of benign and inconsequential missense mutations in the human genome means that the confidence of constraint metrics is drastically lower (ref 36). Hence, the negative predictive value is low, i.e., not finding constraint for missense mutations does not provide meaningful confidence that constraint, perhaps in just specific exons or loci of a gene, is not present.”

We continue to believe that this paper should address constraint of missense mutation, even if the pangenomic data is currently insufficient to demonstrate fully analytically useful metrics. The sentence “meaning that the AD/XLR pCPS genes that were (likely) not constrained for pLoF mutations also showed no difference in missense mutation constraint”, which you mention, underscores the finding that the AD/XLR genes which do not show constraint of LoF variants (where a reasonable theory is that a real childhood cancer risk could be driven by missense variation) also do not show any sign of missense constraint. It is, of course, probable that future data will affect this conclusion, but based on the current evidence this is the relevant working hypothesis.

In this regard, *RET* is not relevant; this gene *does* show constraint of LoF variants, so, if we understand you correctly, the sentence does not address this gene. The pCPS is associated with GoF variants, and as such it is now, as detailed in your point 8 and elsewhere, separated out early for separate analysis. We very much agree that the LoF constraint observed in *RET* is likely caused by non-cancer phenotype. But this does not change the fact that the gene, which is linked to risk of childhood cancer, is clearly constrained, albeit likely unrelatedly.

We have altered the relevant text to reflect these points.

8. Related to number 7, the point that some of the CPS genes, e.g. HRAS, are well documented to have gain of function variants as the disease mechanism. This is not a surprising finding and much more of a confirmation of the validity of the analysis. This data should be raised much earlier in the

paper and their lack of constraint described as essentially a control for the analysis not a new finding.

Thank you for pointing out this very insightful and important observation. We fully agree and have altered the structure of our paper to extensively accommodate this suggestion, which we see as a great improvement in terms of clarity; thank you!

9. The description of the analysis for *ELP1* and *GPR161* is very interesting as these are more recently described genes and the field is trying to understand the appropriate surveillance for these findings. Isn't the most likely reason for the lack of constraint for both genes low penetrance. Why is positive selective pressure described for *ELP1* and not *GPR161*? Please clarify this analysis.

Thank you for complementing this section of the manuscript and for highlighting the need for improving our understanding of these more recently identified CPS risk genes.

And yes, we fully agree; Occam's razor favors low penetrance as the explanation as that introduces the fewest new assumptions (positive selective pressure has not been established for any variants which cause high risk of childhood cancer). Following up on your input, we have added a sentence to the discussion of *ELP1* to further clarify this in the revised manuscript. And as you mention, the argumentation is the same regarding *GPR161*, i.e., positive selection could be an explanation, but this seems unlikely. We hope that this sentence communicates this sufficiently; "Hence, the argumentation mirrors that of *ELP1* above, and again low penetrance appears to be a likely explanation for the lack of evidence of pLoF constraint."

10. The description of the association of *MSH2* with constraint is very interesting and as noted is consistent with the recent meta-analysis by Kratz et al., JNCI, 2022. However, it is likely that the accumulation of pediatric cancer studies used in the analysis here strongly overlaps with the studies put together in the Kraft et al paper to complete their analysis. The authors should carefully review these two cohorts to confirm that there is no redundant data. This is particularly important as several recent studies of Lynch syndrome diagnoses (not CMMRD) in pediatric cancer (not cited by the authors) did not show any prevalence for *MSH2* as the causative gene and patients are described with all of the MMR genes – Fiala EM et al., Nat Cancer, 2022, Scollon, S et al., Ped Blood Cancer, 2022, and Macarthur, TA et al., J Pediatr Surg Case Rep, 2022.

Thank you for your support to this observation. It is an important point which strikes at the heart of the utility of mutational constraint as an analytical metric of distinct value. For clarification, the findings that *MSH2* is constrained for LoF variants is entirely independent of the data presented by Kratz et al., JNCI, 2022. *MSH2* is *only* selected as potentially interesting (along with *DIS3L2*) because it, as a conventionally biallelic childhood cancer risk gene, shows constraint in the germline exomes and genomes of adults included in gnomAD v2.1.

We definitely agree with your point in regard to the statistical test specifically comparing pediatric cancer study findings to gnomAD. We have updated the wording so that the presence of the overlap is clear. We feel that showing that the difference in constraint exists in relation to gnomAD is of additional value, as Kratz et al., JNCI, 2022 did not discuss any of their findings in relation to constraint, even though it supports their conclusions (in our opinion further underscoring the value of this paper as a future reference in such discussions).

As response to the important final sentence of your point; yes, we are familiar with and include several studies reporting on heterozygous pathogenic mutations in MMR genes, yet, unlike the study by Kratz et al, the three studies by Fiala, Scollon, and Macarthur et al. do not have a control cohort and as such the studies do not meaningfully address causation.

11. The historical axes figure and discussion seems unnecessary. Discovery is often a function of methodologies available at the time. The earlier discoveries often required large families which one would argue are less likely to be constrained genes. Exome sequencing of trios looking for de novo variants (likely to be constrained genes) developed later. Penetrance is another important aspect of gene/disease discovery and would favor more constrained genes. Certainly, all discovery pipelines now uniformly look carefully at constrained genes for severe pediatric phenotypes (whether cancer or others) so this again seems like an unnecessary section to an already long paper.

Thank you for this input and for adding to the discussion which led to text edits underscoring these points. We assume you are referring to the right panel of Figure 4 (Figure 3C in revision). In our view, visually, this data does not seem to support that historical timing of discovery (and hence any derived linked methodologies) favor a particular level of LoF variant gene constraint. Also, you write that all discovery pipelines certainly include constrained genes. We were as surprised as you to see that this is indeed not the case; we were unable to find a single paper that meaningfully includes evolutionary constraint as a factor in gene discovery related to childhood cancer risk — this served as the motivation for our investigations. We see this point as fundamental to the relevance of this work. The gene examples mentioned in relation to Figure 4, incl. *ETV6*, *FBXW7* and *DIS3L2*, have all been presented in papers which do not mention mutational constraint whatsoever, although for each of these genes, the observed constraint supports their hypotheses; a cancer syndrome highly penetrant for childhood (i.e., pre-reproductive) cancer. The same may be said of *ELP1* and *GPR161*, where, a complete lack of discussion of mutational constraint arguably led to an overestimation of clinical relevance for childhood cancer, which could have been tempered by a proper discussion of this data. We certainly agree that other subfields of genetics have adopted mutational constraint in their interpretation of both known and novel genotype-phenotype correlations, but this has not penetrated into the field of pediatric cancer germline genomics - we believe our paper supports that it should.

Minor:

1. Figure 3 needs appropriate legends (what do the colors stand for) and definition of the categories for the reader to follow. What is the X axis designating? Similarly, Figure 6 A is missing the genes names on the X axis.

[Full review comments are provided, without modification, in *black italic font*, with our response in **blue font** directly below]

Thank you for pointing out these issues. We agree and have updated the figure according to this and journal guidelines.

Reviewer #4 (Remarks to the Author): Expert in cancer genetics and evolution, and population genetics

Stoltze et al. aimed to quantify the selective constraint on genes associated with pediatric cancer predisposition syndrome (pCPS) in their paper. Given that most pediatric cancers are linked to poor survival rates, which nearly preclude reproduction, genes with mutations that increase cancer risk during childhood are expected to be under strong selective constraint. By focusing on the loss-of-function observed/expected upper bound fraction (LOEUF) metric reported in the gnomAD database, the authors indeed found a severe depletion of unique loss-of-function (LoF) variants in pCPS genes observed in adult humans than neutral expectation. For some of these pCPS genes, they also observed seemingly enrichment of LoF mutations in children with cancer. The authors then discussed a few autosomal dominant pCPS genes that lack evidence of constraint, as well as two autosomal recessive pCPS genes that show strong evidence of constraint, which suggest false association to childhood cancer risk or incorrect mode of inheritance.

Thank you for this succinct recap. We are very grateful that you have taken the time to critically evaluate our work, which we found immensely valuable.

It is a nice idea to examine the selection constraint of pCPS genes, and I am largely convinced these genes are subjected to quite strong selective pressure. However, it is worth noting that the LOEUF metric only measures selective constraint on LoF variants and has several limitations, including incomplete and biased power across genes. The biased power suggests that it is crucial to match different sets of genes to avoid potential confounders, while the incomplete power means that although genes with low LOEUF scores are highly constrained, but not all constrained genes necessarily have low LOEUF scores. As a result, it requires caution to conclude that certain gene lacks selective constraint. In addition, the method that the authors used to quantify enrichment of mutations in children with cancer is highly problematic, so I am skeptical about findings from this analysis. Lastly, the methods are not sufficiently described in a couple of places, and the figures could be improved. My specific comments are outlined below.

We have added a point-by-point to your comment below.

Major points:

1. Limitations of using LOEUF score for measuring selective constraint

First, this metric is purely based on loss-of-function variants, so it only detects signal when complete loss of one copy of the gene (or 50% reduction in gene expression) is deleterious, but this metric cannot measure selective constraint on gain-of-function mutations or other types of mutations. This is a particular concern in the analysis of cancer driver genes, because cancer driver genes include both tumor suppressor genes, the loss of which leads to cancer, and oncogenes, the activation of which leads or predisposes to cancer. Can the author break the pCPS genes into two classes, tumor

suppressor genes and oncogenes, and analyze them separately? My expectation is that tumor suppressor pCPS genes would be more enriched for LOEUF-based constrained genes compared to random gene sets, although the oncogenes may not be (this does not mean that the latter group is not constrained; it just means that these genes are not significantly depleted for LoF variants). Along this line, this expectation fits the observation for SAMD9 and SAMD9L, pathogenic variants in which are gain-of-function. I hope the authors could discuss the distinction between oncogene and tumor suppressor genes earlier.

This is a really important point, and we agree fully. We have restructured the paper so that pCPS driven by GoF variation (relevant for nine genes; equivalent to oncogenes) is now treated separately from the beginning. We feel that this greatly improves the clarity of the work, yet, the conclusions drawn remained unchanged after this implementation.

Second, LOEUF has different powerful in detecting signal of selective constraint from gene to gene. For example, one specific bias is much reduced power for short genes, because it is challenging to detect a significant depletion from expectation when the expectation is already low. Comparing Fig2 A and B, it seems pCPS genes tend to be associated with narrower confidence intervals compared to the random set, possibly because pCPS genes are enriched for long genes. In addition to gene length, other genomic features of a gene may also bias the LOEUF's detection power. Given this bias, it would be nice if the authors can match for gene length and other features when drawing random genes to compare with pCPS (for example, analysis corresponding to Fig 2AD). Similar concerns apply to the O/E ratio of missense mutations shown in Fig 7, the comparisons between high vs low risk pCPS genes and between high vs. low mutation rate pCPS genes, and comparison between adult and childhood pCPS genes.

This is an excellent point, and we fully agree that this will improve the comparative analysis between pCPS genes and all other genes. Following up on your input, we have rerun the analyses using random sampling that accounts for gene size. All relevant figures and derived statistical metrics have been updated to reflect this change, and a description of the gene-size matching has been added to the methods in the revised manuscript.

Lastly, LOEUF relies solely on number of unique LoF variants observed in the same but ignores the allele frequencies of the observed variants, so it has little to no power in detecting selective constraints on variants with recessive fitness effects, the signal of which is lower allele frequency than neutral expectation. Accordingly, the statement in lines 102-104 is inaccurate: technically, genes where LoF variants have completely recessive effects are also constrained, which is manifested in lower frequencies of deleterious mutations, although this depletion is likely not picked up by LOEUF.

Yes, we certainly see your point and have hedged the relevant language, so it is not so absolute in the revised manuscript. In practice, for a truly AR genotype, only the often exceedingly rare q^2 state of the Hardy-Weinberg equilibrium will be under selective pressure (and only as strongly as dictated by

the reproduction-limiting penetrance of the reproduction-limiting phenotype). Whether constraint with regard to allele frequencies exists for AR genotypes is not obvious to us and is likely to be mostly unknown. It is not atypical for common pathogenic AR genotypes (i.e., founder variants) to have a heterozygote advantage — presumably driving them to become common in the first place. This, coupled with the fact that the much more common pq state is otherwise free to spread in the population as a neutral variant is very likely to limit the chances of an AR gene becoming discernibly constraint (even with exceedingly high power), which indeed mirrors our empirical findings as well as those of others.

2. Estimating number of LoF variants in children with cancer by extrapolation

The authors do not explicitly explain how the extrapolation was done, but based on the numbers provided in Fig 5, it seems they assumed that the number of LoF variants observed would be proportional to the sample size (i.e., the number of children) and compared the scaled number to the expectation and observation in adults. This extrapolation method is incorrect, because both the expectation and observation (in adult human) reported by gnomAD are the number of unique LoF variants, which does NOT scale with the sample size. In fact, this number is expected to increase slower as the sample size grows larger, which can lead to the apparent “enrichment” of LoF mutations in children with cancer when the number is simply scaled up. Instead of extrapolation, the author can look at dataset with WGS data available (such as UK10K), down-sample the adult data randomly (multiple times) to match the number of children, and ask how often the number of unique LoF observed in adult samples is lower than observed in pediatric cancer patients. Alternatively, the authors can compare the average number of LoF per individual between adults in gnomAD and children with cancer, by summing the allele frequencies of all LoF variants observed in that group. This approach is unbiased with regard to the sample size, but the authors need to account for uncertainty properly especially for genes with small sample size. If the results still hold, this is evidence for “evolution in action”, but it is still an over-statement that the increased cancer risk is what is driving natural selection (lines 155-157).

Thank you for raising this point. We agree that the extrapolation is not sufficiently explained in the original submission. Also, we realized that in the text, there was an erroneous number based on linear extrapolation, which is sure to have added to the confusion. We hope you will accept our apology. This has now been fixed so the revised manuscript text matches Figure 5 (Figure 4 in revision). The calculation is made assuming a parabolic relationship (figure text in Figure 5 and methods which also point to the relevant section of the code where this is calculated in R, as well as give an example to clarify the relationship assumed). We, of course, fully agree that, since constraint speaks to the number of distinct genotypes, assuming a linear relationship is nonsensical. Using a parabolic extrapolation, as we have done, assumes that the “number [of distinct LoF variants] is expected to increase slower as the sample size grows larger”, exactly as you describe. It is, thus, our impression that your comment might reflect a misunderstanding caused by lack of clarity in our original method and writing. As an example, based on 9 observed variants in 4,833 children with cancer, the erroneously presumed linear extrapolation would yield an extrapolated number of distinct LoF variants in *APC* of 186 [$9 \times (100,000/4,833)$], yet, the presented extrapolation is calculated as 41

[$9 \times \sqrt{100,000/4,833}$]. Reviewer #1's comment led to us accounting for an overlap in the reported pancancer studies, which alters the denominator, so that the recalculated extrapolated number for *APC* now is 37. We have added clarifying data to Figure 5 (Figure 4 in the revision) where we also, based on your excellent suggestion performed downsampling for each gene included in this analysis and added plots to the revised Figure with these results. We feel that this now clarifies what is actually observed vs. our own extrapolation and greatly improved clarity of this part of our study.

3. Selection constraint driven by childhood cancer risk

The authors found that pCPS genes with neoplasm as the only known phenotype are still highly constrained and concluded that “stand-alone early life cancer risk can drive natural selection” (lines 139-140; similarly in lines 185-186). However, this conclusion is premature, because mutations in these genes can still be pleiotropic, causing higher cancer risk and other weak phenotypes simultaneously, and the subtle, subclinical phenotypes may lead to severe reduction in fitness (for detailed discussion of phenotypic vs. fitness effect, please see Fuller et al., 2019 Nat Genet). In fact, most constrained genes are not associated with disease or fertility problem, suggesting that subtle, clinical phenotypes can contribute to fitness loss (Gardner et al 2022 Nature).

We very much agree with this insightful comment. Accordingly, we have tempered the relevant language used to underscore that our analysis merely suggests that a relationship between variants in a gene causing childhood cancer and that gene being constrained exists. It is highly biologically plausible that there is a causal relationship; in the referenced work this is also highlighted as a necessary feature of equating constraint and lack of fitness. With the adjustments made to the revised manuscript, we believe that this observation, based on well-documented genotype-phenotype correlation with a direct mechanism for the selective pressure, is adequate. Naturally, it is based only on the data available, and as such, it stands to be further evaluated as pangenomic data availability scales and any undiscovered pleiotropy is uncovered.

4. Discussion of *DIS3L2* and *MSH2*

In the abstract, the authors wrote that monoallelic LoF mutations in these two genes may increase childhood cancer risk, but this conclusion was not explicitly stated in the main text of the corresponding section. The presence of selective constraint alone is supporting evidence but not sufficient, as these genes may be truly recessive with regard to cancer risk, but heterozygotes carrying one defective gene copy may have other weak, subclinical phenotypes that reduce fitness subtly, leading to selective constraint (for example, even 5% reduce in fertility is considered evolutionary highly deleterious). A more compelling piece of evidence is enrichment of heterozygotes in children with cancer, but why does *PMS2* lacks selective constraint despite being enriched in children with cancer? Also, how do the authors reconcile the observations that heterozygous *DIS3L2* LoF mutations were seen in 5 out of 126 children with Wilms tumor but none observed in the 3045 patients across 11 pediatric pan-cancer studies? How to interpret the observation of only one overlapping LoF mutation between children with WT and adults? The authors need to state their argument, explanation, and conclusion more clearly in this section.

These are excellent points, and we have made several edits to the revised manuscript based on your comments and suggestions. As we agree that a final conclusion cannot be made in this regard, we altered the language in the abstract to reflect that childhood cancer risk *may* be an explanation of the observed constraint in *MSH2* and *DIS3L2*. Importantly, and as you point out, because an enrichment is seen in children with cancer, variation in the gene being causative of childhood cancer is a fair explanation and introduces fewer assumptions than an uncharted infertility phenotype, although we, of course, agree that this cannot be ruled out. With regards to why no constraint is observed for *PMS2*, we can think of several possible explanations; firstly, and most importantly, the enrichment of pathogenic *PMS2* in children with cancer found by Kratz et al., JNCI, 2022 was observed only in comparison to their smaller German control cohort of 27 501 individuals, yet, in contrast to *MSH2*, no difference was seen for *PMS2* in comparison to gnomAD. This could suggest a bias in the German controls. This is why we write that “*MSH2* was [the only classically adult-onset CPS gene] *consistently* associated with childhood cancer risk”. Secondly, *PMS2* has a pseudogene which greatly affects the call-rate of pathogenic variants and obfuscates interpretation. After performing germline WGS sequencing in a cohort of children with brain tumors which we recently published Stoltze et al., Neuroonc., 2022, we had to do long-range sequencing to discover that two children with the same *PMS2* variant, originally classified as pathogenic, were actually a benign variant in the pseudogene. This is well-known, yet not discussed by Kratz et al., JNCI, 2022, and the size of this phenomenon is somewhat unknown and may also result in separate differences across ancestries. For these reasons, we did not deem it necessary to add to an already lengthy discussion with regard to this observation. Of course, a low penetrance of childhood cancer is not incompatible with *PMS2* not being constrained, but we believe that deeper discussion of this seeming incongruence should be reserved for a time when heterozygous variants in the gene have been more definitively linked to childhood cancer risk, if this is indeed the case. Finally, with regards to *DIS3L2* variants not being reported in the 3,045 cases where the gene was on the chemical/in silico panel; this could, of course, be due to the fact that 1) the genotype truly causes monoallelic childhood cancer risk, but is rare, 2) the observations in the Dutchstudy could be an error or, perhaps just, ancestry-dependent, yet we do not think that such speculations are warranted based on the data. This is because many of the pancancer studies, which all pre-dated the paper describing the observations regarding *DIS3L2* mutations in children with Wilms tumor, often did not report heterozygous carrier statuses for genes, like *DIS3L2*, understood at the time to be AR. Hence, reporting bias may exist and further validation of the *DIS3L2*, outside the one country which has reported it, is needed. This latter point has been added to the relevant section of the revised manuscript following your input. Finally, we have added text clarifying that the relative lack of overlap in mutational spectra between children with cancer and the population could mean that LoF variants in children with cancer have a distinct spectrum but that such a theory requires further substantiating data.

5. Questions regarding methods

1) Fig 1: For the definition of “Likely constrained” and “Likely not constrained” genes, is there additional criteria in addition to $LOE < 0.35$ vs $LOE > 0.35$? If not, is it meaningful to classify genes into these two categories purely based on the LOE value, for genes with large confidence intervals?

Yes, we believe that this best reflects the current understanding of the gene constraint and that the addition of ‘likely’ to the label appropriately indicates that further data is needed to make a more confident conclusion. That being said, which cut-offs and labels to use is indeed debatable, and we selected one way of categorizing, and made sure that this was fully transparent.

2) In the simulation of 1000-year-survival change of mutations in pCPS genes, is it assumed that the cancer phenotype, once manifested, is incompatible with survival/reproduction? Is this true for all or most types of childhood cancer?

Very valid point; spontaneous remission has rarely been described, mostly in unusual circumstances (PMID: 18719616). Furthermore, adolescent cancer patient may occasionally live long enough to have an offspring. Ultimately, we deemed these scenarios to be of insignificant consequence, and thus this core assumption stands as stated in the text. We find the assumptions to be largely fair as emphasized by the very high mortality rates in pre-Western-medicine history and low- and middle income countries, when contemporary therapy cannot be provided.

3) How is the weighted ratio on the y-axis calculated in Fig10A? How many unique genes are involved for the 229 and 485 pathogenic mutations found in adults and children with cancer? Why is it meaningful to divide the genes in four groups, when the distinction between the two middle groups is somewhat stochastic?

Thank you for highlighting this. Upon deliberation, we think that the label ‘weighted ratio’ is a misnomer, as it is in fact just a simple ratio. We have updated the relevant figures’ axis labels.

4) How do the authors translate the 1,047 cancer risk SNPs into 722 genes, especially for SNPs not found in gene body?

We used the genes reported by the discovery studies and have taken care to write that the SNPs were merely “in proximity to” the genes investigated.

Minor comments:

In lines 50-53, the five bullet points should be all verbs so that are parallel to each other.

Agreed; corrected.

Fig6: it is not obvious to me what this figure is meant to show and why it is cited in the section “pCPS genes with monoallelic cancer risk as the sole phenotype show constraint”.

The figure (Figure 5 in the revision) was meant to illustrate the current level of insight into the variants that cause childhood cancer — for instance that many genes are only very rarely (or not at all) mutated even across the many reported children with cancer. We believe that this is of interest to people within the field of pediatric oncology. We have decreased the real estate taken up by the

[Full review comments are provided, without modification, in *black italic font*, with our response in *blue font* directly below]

figure and combined it with the figure on missense constraint and finally added text to highlight that some of the not constrained pCPS genes had only missense variants reported in the pancancer studies — yet these do not show any signs of missense constraint based on the current data. We hope that these changes clarify the relevance of the now reduced figure.

Fig 9: Is the order of genes in Fig 9A meaningful? If not, can the authors sort the genes by their LOEUF score to avoid crossing of the arrow heads? What does TUM (corresponding to triangles) stand for? What is the interpretation of results shown in Fig 9B?

Excellent suggestion; yes, this is certainly much better, and the figure has been changed and legend/text updated as suggested. For 9B (7B in the revision), the interest stems from the fact that the gene is both LoF constrained and shows enrichment of LoF variants in a childhood cancer — your question is similar to your point #4 above, please cf. the changes made to the text under this response.

For a manuscript of this length, there are too many figures. I would suggest the authors remove or combine some of them (5-6 figures would be ideal).

We agree with this point and have reduced the number of figures to 8, please cf. major points at the beginning of this document.

Line 359: the word “both” should be removed, as three types of biases are discussed.

Thank you; fixed.

Line 388: “form” should be “from”.

Yes, that makes way more sense.

Reviewer #5 (Remarks to the Author): Expert in bioethics of reproductive technologies, prenatal selection and screening, and genomic medicine

This paper focuses on the population genetics of childhood cancer, and makes some remarks about possible interventions and their implications for the human gene pool. Briefly, the authors suggest that the link between genes associated with childhood cancer and ‘constraint’ (an unexpectedly low rate of mutation) is explained by natural selection. Developing treatments that cure the phenotype of childhood cancer, thereby allowing children carrying those genes to live to sexual maturity and pass their cancer-causing genes on, will act against natural selective pressure, resulting in those mutated genes eventually becoming more common in the population. To counterbalance this effect they suggest the use of a variety of prenatal screening methods (population based screening, preimplantation genetic diagnosis of in vitro-produced embryos, and “other family planning interventions”) to ensure those future individuals still carrying the mutated genes do not appear in the population. They argue that “Medically deselecting embryos with variants in highly constrained genes effectively mirrors the selection process of nature evident in the human gene pool today.”

Thank you for this accurate recap. We are very grateful that you have taken the time to critically evaluate our work, which we found immensely valuable.

In other words, those children who are already born will be treated to enable them to survive their cancer, or other genetically related disease; but we will act (or at least, offer people the opportunity to act) to prevent further individuals from being born and going on to develop a (presumably now treatable) cancer.

It is incontrovertible that curing paediatric cancer is a good thing. There is more ambiguity over the moral rightness of intervening prenatally to ensure that people with particular traits, like paediatric cancer, are not born. Bioethicists specialising in disability have developed a critique of prenatal selection against disabling conditions, arguing that at times it can be nothing more than discrimination against people with disabilities that are perfectly compatible with a good and flourishing life. But on the whole they aren’t making this argument about conditions as severe and potentially lethal as paediatric cancer.

We fully agree with these expanding views on our original submission, however, these fit thematically with the section which, upon much deliberation, has been removed; please see point #4 at the beginning of this document as well as the final our final response below.

Nevertheless, there are still some major ethical considerations to be discussed. For one thing, prenatal identification and selection, by whatever method, effectively becomes a built-in part of the management of certain paediatric cancers (for the survivors once they are adult); this may be the morally right thing to do for the individual, but has implications for the routinisation of selection for the community as a whole.

Second, in contemporary biomedicine there are many situations in which a proxy – such as a gene locus – becomes a marker for the thing itself. The fact that the marker is some steps away from the phenomenon it stands for can easily slip out of sight. In prenatal genetic testing and screening, an allele stands for a predicted experience of suffering or disadvantage that we consider severe enough to call ‘disability’ and to warrant termination of pregnancy. For obvious reasons, in this situation use of the gene locus as a proxy is unavoidable as it is the only information we have; but the extent to which it’s an accurate predictor of future suffering or disadvantage depends on numerous factors, including social and environmental ones. The paper suggests that the logic of observing that genes causing paediatric cancer are ‘constrained’ could be reversed: that if genes are found to be constrained then that might be an indicator that their mutated versions are likely to be pathological. As the authors paper note, there are around 3000 human genes that show constraint but the majority of these are not (yet) linked to pathology. But it is not inconceivable that a gene showing mutational constraint will come to be considered suspect in itself even if no link with disease has yet been shown, and that its presence will be assumed to raise the risk profile of the carrier. There should be some caution about a growing reliance on markers that are more and more distanced from experiencing the pathology they are supposed to indicate.

Thank you for bringing up these excellent points. We have added the concluding paragraphs of the paper to underscore this, as suggested.

A final point to think about: the rationale the paper gives for using prenatal selection in these cases is essentially about avoiding long-term, unwanted changes in the human gene pool. In stark contrast, most population prenatal screening today is offered on the basis of the lives of individuals: enhancing individual reproductive autonomy and avoiding suffering (of the future child). Screening programs strenuously avoid the population-level arguments about the quality of the gene pool reminiscent of the historical thinking and practices that today would be condemned as overtly eugenic. Whether or not prenatal reproductive selection can justifiably be called ‘eugenic’, as some ethicists and activists do, is a highly contested point and not one I’m going to address here. But it is important to remain alert to the risks, both actual and perceived, of a scientific advance that fundamentally shifts the moral framework within which people consider reproductive choice.

Thank you again for taking the time to review our work. We read your comments with immense interest, and we fully agree with all of your observations. Indeed, we were glad to see that our work elicited such reflections, which we consider, in part, to be the foundation for the exploration and publication of this paper focused on the Darwinian consequences of childhood cancer. Following both your comments and similar comments made by Reviewer #3, we asked Nature Communications for editorial input on how to best address these aspects. The consensus was to entirely remove the section previously found under subheader “Reverting the effects of childhood cancer on the human gene pool”. We are convinced that this subject is worthy of scientific discussion, however, it is clear to us that it should be reserved for a paper with this as its core focus — allowing comprehensive deliberations on this ethically complex matter. We are confident that our manuscript, now revised based on your expert guidance, should provide an indispensable reference for such a discussion.

REVIEWER COMMENTS

Reviewer #1 (Remarks to the Author):

I much appreciate that the authors followed my suggestion and accounted for a 259-patient overlap between two major pediatric pancancer studies (Zhang, NEJM2015 & Gröbner, Nature2018). Glad to read that this change provided an important improvement in accuracy, but did not have any impact on the conclusions drawn.

Further, they have rerun the analyses using random sampling that accounts for gene size, which is an improved methodological approach and makes their results more robust.

In summary, I agree with the authors that the implemented changes based on all of the reviewers' comments have greatly improved the clarity and technical aspects of the manuscript and I'm looking forward to seeing this study published soon.

Reviewer #3 (Remarks to the Author):

The authors have substantially revised the manuscript and responded in detail to my prior review. I appreciate the effort that went into revising the manuscript, removing problematic sections, adding reference to other groups that use constraint in medical genetics and updating the discussion. Overall, I think the authors make many important points and illustrate them with a wide variety of figures. I have noted specific corrections/suggestions below.

Points to correct/update:

Figure 1 (Survivorship) – there is an important error in the top and bottom panels of the figure – “Damage observed (intolerance can be inferred)” should be “Damage observed (tolerance can be inferred)”.

Same thing for the bottom panel. Also for the bottom panel be clear that you are talking about loss of function or deleterious mutations. There are many mutations that don't have any impact on evolution. That is why we use different measures for Loss of Function versus Missense variants.

The Gain of Function genes, e.g. RET, also have loss of function phenotypes that could be evolutionarily constrained. For example, LOF RET alleles cause Hirschsprung's disease which is likely to be constrained and should potentially be mentioned in the discussion as the GOF data is not discussed further.

Lines 148-150 unfortunately use the word "only" known phenotype of neoplasms. They still include SMARCA4 and SMARCB1 as genes that only have childhood cancer. That is true of loss of function variants (for those two genes) but not missense or inframe deletions. DICER1 also demonstrates macrocephaly. Perhaps best to say that LoF variants in these genes are predominantly associated with neoplasms.

Figure 4 – the graphics are really hard to read on a large, very high resolution monitor. Defer to the editors but I would really think of other approaches to this figure.

Figure 5D – Since 5C is constraint for missense variants and 5D is grouped by LOF scores it would be important to be clear on the legend when are you talking about constrained for LOF versus missense. I'm not sure I follow it on Figure 5D.

Line 314-315 – Most mRNA from transcripts with LOF variants undergo nonsense mediated decay, so you don't expect them to necessarily cluster within the gene. Many genes like RB1 also don't show clustering.

Line 352-355 – is incorrect and needs to be updated. Parsons et al., JAMA Oncology 2016 – reference 43 – is one of the 11 studies included in this study of pediatric CPS studies. That paper does specifically note that they identified one patient with monoallelic DIS3L2 variant in a patient with Wilms tumor and the tumor showed completed LOH at the DIS3L2 locus.

In the discussion about MSH2 being constrained it may be worth noting that there 3 recent papers demonstrating that Lynch syndrome (monoallelic) variants in the 4 mismatch repair genes are enriched in pediatric cancers with a particular enrichment in pediatric brain tumors which are highly lethal. However, the penetrance may be below what is detectable by the constraint approach used here.

Reviewer #4 (Remarks to the Author):

The authors have partially addressed my concerns from last round, but substantial revision is still needed. In addition, there are numerous small

Major points:

1. Biased performance of LOEUF score for different genes

Although the authors acknowledged the differential power of LOEUF for detecting selective constraint for different genes and made an effort to correct this by making comparison to size-matched genes in some analysis, it seems they do not fully appreciate the underlying causes of this biased power. As a result, some issues still remain in the manuscript. For instance, in the comparison of pCPS genes with low vs high de novo mutation rates, for reasons similar to those for short genes, it is harder to detect significant depletion of LoF variants for low mutation rate genes. Therefore, the comparison between low- and high-mutation rate pCPS genes is simply unfair and does not provide sufficient support for the statement in lines 227-229. Similarly, the comparison of LOEUF scores between pCPS genes with low and high penetrance can well be confounded by differences in gene lengths and/or de novo mutation rate. I hope the authors could carefully reconsider potential confounders in all comparisons of LOEUF scores between gene sets throughout the paper.

2. Number of LoF variants in children vs. in healthy adults

My concern largely persists regarding the analysis shown in Fig 4. I appreciate that the authors explain how they calculate the expected number by extrapolation, but the assumption of a parabolic relationship between the number of unique variants and sample size is poorly justified. Such a relationship is only expected for neutral variants under a constant-sized population but fits poorly for LoF variants in humans due to recent super-exponential population expansion of human populations and the deleteriousness of variants (which leads to enrichment in low frequencies). Consequently, the results shown in Fig 4 are still unreliable. In contrast, the downsampling approach is more trustworthy and should be focused on in the text and figure. Given that the gnomAD dataset does not provide individual-level data, the authors generated an approximate dataset by randomly assigning the observed number of variants to 125,748 rows (each representing an individual). However, this random assignment ignores linkage disequilibrium among variants and hence may not represent the actual distribution of variants among individuals. For best results, I recommend the authors carry out subsampling from actual individual-level data such as the UK10K. Alternatively, the authors need to explicitly describe the methods used in main text and clearly lay out the assumptions and caveats of their approach. I also recommend the authors carry out the second analysis that I suggested (e.g., compare the average number of LoF per individual between adults and children with cancer, by averaging the allele frequencies of all relevant LoF variants observed in each group). Lastly, to make sure there isn't any technical problems with the genome-sequencing or variant-calling in the pediatric cancer studies, it would be helpful to repeat the analysis for a random set of unconstrained genes as a negative control.

3. Simulation of childhood cancer risk variants

Several key assumptions and parameters need to be clarified. First please clarify the assumption about dominance of the pathogenic variants in the simulation. I presume fully dominant effect is assumed but this needs to be explicitly stated in the text. Second, the population size (e.g., number of individuals) simulated needs to be stated in the text. The authors wrote “an infinite population” in their github code, but this cannot be true, because there is no genetic drift in a truly infinite population, and no variants would ever be lost. Overall, the simulation seems like an overkill to illustrate the point that “even highly incomplete penetrance will lead to mutational constraint”. Given complete lethality and full dominance, the penetrance level is equivalent to selection coefficient against the variant. With a finite population size of $N_e=20,000$ (the current estimated for human), a selection coefficient of 0.1% (which is equivalent to 0.1% penetrance under this model) is already considered to be strongly deleterious. Therefore, the simulation results can be largely replaced by simple back-of-envelope calculation/explanation.

4. As I explained in my review last time, although there is evidence that pCPS genes are subjected to strong selective pressure, it is unclear if the selection constraint is fully driven by childhood cancer risk (obviously childhood cancer place selective constraint, but it is questionable whether other subclinical phenotypes also contribute). Therefore, I suggest modifying the title to something like “the evolutionary constraint of genes underlying childhood cancer risk”.

5. Choice of statistical test

It is inappropriate to use t-test to compare LOEUF score, gene length or other metrics of two groups of genes, as these metrics are usually not normally distributed; instead, please use a nonparametric test (e.g., Mann-Witney U test, or K-S test) to compare the distributions. This applies to the numerous places in the paper (e.g., in lines 100-101, 118-119, 121, 123-124...).

Minor points:

1. “Selective mutational pressure” in the abstract can be replaced by “selective pressure”.
2. The word “put” is unnecessary in line 57.
3. The two sentences in lines 59-61 and 61-63 are redundant and repetitive. Please remove one.
4. Figure 1, the top panel describing the survival bias with WWII bombers is unnecessary; the authors can keep the genome illustration (the lower half of Fig 1A) if wanted. In addition, the label “Likely not constraint [LOELF<35%]” in Fig 1B is incorrect; I believe it should be “Likely not constraint [LOELF>35%]”. In the legend of Fig 1B, LOELF and LOEUF should represent the “lower and upper bounds of the 90% confidence interval” instead of “lower and upper fraction of 90% confidence interval”.
5. Please explain how the subset of genes matched to pCPS genes by size was selected.
6. In line 224, “be founded” can be replaced with “newly arise”.

Reviewer #5 (Remarks to the Author):

I have read the revised version of the manuscript and the authors' comments on the revision. The revision follows the editorial recommendation to remove one of the sections. I consider this has dealt with the most contentious points I originally raised. I appreciate the open-mindedness with which the authors have considered my and the other reviewers' comments. I am satisfied that my concerns have been addressed in the revision.

Overview

Dear reviewers,

Thank you for providing us with highly constructive input to our revised manuscript. As an introduction to the detailed point-by-point responses below, we here provide an overview of the most substantial changes made in this 2nd manuscript revision. We are, once again, immensely grateful for your willingness to commit your expertise and time to ensuring the highest quality for our work.

- **Statistical testing.** Statistical tests have been revised to be non-parametric, employing Mann-Whitney U (MWU) in favor of t-test. This change is more suitable for the data, and did not lead to any consequential changes to results or conclusions. Where appropriate, head-to-head comparisons of gene sets have been corrected for gene size.
- **Figure changes.**
 - **Fig 1:** Removal of the WWII bombers analogy, keeping only the panel relating to genomic variation and survival bias. Changed wording. Clarified definitions.
 - **Fig 2:** Updated p-values to reflect test revised using MWU.
 - **Fig 4:** Removed extrapolations and misleading downsampling (see detailed response below). Added new plots based on direct downsampling of gnomAD (individual-level data), and direct (non-extrapolated) data from peds pancancer studies. A Supplementary Fig 3 was added for the sake of completeness showing the same graphs as Fig 4C for all 11 remaining genes (not included in Fig 4C due to figure size/readability).
 - **Fig 5:** Updated p-values to reflect test revised using MWU. Clarified legends.
- **Updated code and methods.** Especially regarding the downsampling data, see also data availability of the main text.
- **Removed simulation.** After deliberation (detailed below) we elected to remove the simulation of de novo mutation's propagation from the article. As with the bomber analogy, this element was purely illustrative with an aim of benefiting broad readership — yet the reviewers have persuaded us that such elements are best left out. This also makes the article more succinct without detracting any novel insights. Code and code references have been updated accordingly.

Reviewer #1 (Remarks to the Author):

I much appreciate that the authors followed my suggestion and accounted for a 259-patient overlap between two major pediatric pancancer studies (Zhang, NEJM2015 & Gröbner, Nature2018). Glad to read that this change provided an important improvement in accuracy, but did not have any impact on the conclusions drawn.

Further, they have rerun the analyses using random sampling that accounts for gene size, which is an improved methodological approach and makes their results more robust.

In summary, I agree with the authors that the implemented changes based on all of the reviewers' comments have greatly improved the clarity and technical aspects of the manuscript and I'm looking forward to seeing this study published soon.

We are immensely grateful to you for your assistance and thank you for this kind conclusion and for offering your valuable time to the improvement of our work.

Reviewer #3 (Remarks to the Author):

The authors have substantially revised the manuscript and responded in detail to my prior review. I appreciate the effort that went into revising the manuscript, removing problematic sections, adding reference to other groups that use constraint in medical genetics and updating the discussion. Overall, I think the authors make many important points and illustrate them with a wide variety of figures. I have noted specific corrections/suggestions below.

We greatly appreciate your acknowledgement of our efforts. Then and now, we strive to make careful and detailed considerations of each reviewer comment, and we hope that our efforts this time will be viewed in the same light. Thank you for contributing your time and expertise in providing this excellent re-review to our work!

Points to correct/update:

Figure 1 (Survivorship) – there is an important error in the top and bottom panels of the figure – “Damage observed (intolerance can be inferred)” should be “Damage observed (tolerance can be inferred)”.

Thank you for bringing this up. In our estimation it is two sides of the same coin; observed data is used to infer tolerance, and, by extension, also to infer intolerance. To clarify this we have altered to in-figure text to read “tolerances can be inferred”. We agree that this is more neutral. More changes have been made to the figure based on comments by R4.

Same thing for the bottom panel. Also for the bottom panel be clear that you are talking about loss of function or deleterious mutations. There are many mutations that don't have any impact on evolution. That is why we use different measures for Loss of Function versus Missense variants.

We agree. We have clarified that these metrics currently apply for loss-of-function variants, by adding it to the in-figure text.

The Gain of Function genes, e.g. RET, also have loss of function phenotypes that could be evolutionarily constrained. For example, LOF RET alleles cause Hirschsprung's disease which is likely to be constrained and should potentially be mentioned in the discussion as the GOF data is not discussed further.

We agree absolutely, the constraint observed in the *RET* gene is very likely caused by the known non-cancer phenotype associated with loss-of-function phenotype associated with LOF variants. Indeed, any of the 9 genes where cancer risk is driven by GOF, which still show LOF constraint must have a severe LOF phenotype associated with it too. Perhaps some are embryological in nature and don't manifest as a detectable disease. Your point has been added to the relevant comparison.

Lines 148-150 unfortunately use the word “only” known phenotype of neoplasms. They still include SMARCA4 and SMARCB1 as genes that only have childhood cancer. That is true of loss of function variants (for those two genes) but not missense or inframe deletions. DICER1 also demonstrates macrocephaly. Perhaps best to say that LoF variants in these genes are predominantly associated with neoplasms.

We agree that ‘only’ is too unilateral to use in biology. We are hesitant to use the term predominantly, and there are a couple of reasons for this 1) is that it indicates that the majority, i.e. 51-100%, of the known risk is cancer; this is true for several syndrome, which at the same time have strong non-cancer phenotypes (e.g. *ETV6*), 2) is that the “quality” of the risk is difficult to gauge; your example with *DICER1* shows this well, does the observation that an absolute 30% more *DICER1* carriers have subtle non-disproportional macrocephaly compared to family-controls (PMID: 27441995) mean that this is the predominant phenotype (30% is greater than the 5.3% childhood cancer penetrance).

With this in mind we have acted on your suggestion and elected to use the terms “prevailing” and “(apparently) isolated”.

Figure 4 – the graphics are really hard to read on a large, very high resolution monitor. Defer to the editors but I would really think of other approaches to this figure.

Thank you, yes, we definitely recognize this point. Reviewer #4 has made comments that have led to a rework of the illustration, and here we appreciated your point and improved readability.

Figure 5D – Since 5C is constraint for missense variants and 5D is grouped by LOF scores it would be important to be clear on the legend when are you talking about constrained for LOF versus missense. I’m not sure I follow it on Figure 5D.

Yes, we agree that this was a source of confusion. We have added clarifying markers to the legends of the **figure 5D** so it is now clear that the grouping/color is by LOF-constraint.

Thank you.

Line 314-315 – Most mRNA from transcripts with LOF variants undergo nonsense mediated decay, so you don’t expect them to necessarily cluster within the gene. Many genes like RB1 also don’t show clustering.

This is very true, the argument is not essential, and hence we have removed it.

*Line 352-355 – is incorrect and needs to be updated. Parsons et al., JAMA Oncology 2016 – reference 43 – is one of the 11 studies included in this study of pediatric CPS studies. That paper does specifically note that they identified one patient with monoallelic *DIS3L2* variant in a patient with Wilms tumor and the tumor showed completed LOH at the *DIS3L2* locus.*

Thank you very much for alerting us to this. We have reviewed the data import and have identified the source of the problem. Previously we only imported eTable 9 from Parsons et

al., JAMA Oncology 2016 (cf. what was previously **Supplementary data 6**). This table was titled "Diagnostic Germline Findings Related to Patient Phenotype (expanded)", and we missed the fact that eTable10 also includes relevant reports on germline findings, including the heterozygote *DIS3L2* variant you mentioned. We have updated the supp. to include data from this (cf. **Supplementary data 6** in the revised submission, which compiles eTables 9+10 as indicated in the reference sheet, **Supplementary data 1**). After this change we have made the required updates to the manuscript.

In the discussion about MSH2 being constrained it may be worth noting that there 3 recent papers demonstrating that Lynch syndrome (monoallelic) variants in the 4 mismatch repair genes are enriched in pediatric cancers with a particular enrichment in pediatric brain tumors which are highly lethal. However, the penetrance may be below what is detectable by the constraint approach used here.

Absolutely, we have added in more references on this point.

Reviewer #4 (Remarks to the Author):

The authors have partially addressed my concerns from last round, but substantial revision is still needed. In addition, there are numerous small

We are grateful that you have again taken time to review our work carefully. Below you will find our point-by-point response where we have done our utmost to either rectify shortcomings or made them explicit in the text. Your introductory text appears to have been cut short in the transcript we received, but we assumed that the points requiring our attention were all below.

Major points:

1. Biased performance of LOEUF score for different genes

Although the authors acknowledged the differential power of LOEUF for detecting selective constraint for different genes and made an effort to correct this by making comparison to size-matched genes in some analysis, it seems they do not fully appreciate the underlying causes of this biased power. As a result, some issues still remain in the manuscript. For instance, in the comparison of pCPS genes with low vs high de novo mutation rates, for reasons similar to those for short genes, it is harder to detect significant depletion of LoF variants for low mutation rate genes. Therefore, the comparison between low- and high-mutation rate pCPS genes is simply unfair and does not provide sufficient support for the statement in lines 227-229. Similarly, the comparison of LOEUF scores between pCPS genes with low and high penetrance can well be confounded by differences in gene lengths and/or de novo mutation rate. I hope the authors could carefully reconsider potential founders in all comparisons of LOEUF scores between gene sets throughout the paper.

Thank you for bringing forth this important point which led to an overhaul of our testing bearing this in mind. In the highlighted section (preceding ll.227-229), we have taken great care to hedge these analyses, stating “reliable estimates of penetrance associated with most pCPS genes remain scarce” and “[...] this, undoubtedly, somewhat biased observation [...]”, yet we agreed with your points. Firstly, we have revised to employ a model accounting for both gene size and the tested variable, and have updated the text accordingly. Secondly, in rephrasing ll. 227-229, we added text to highlight that there may be crucial confounding which we have not corrected for in these analyses – and we generally toned down language/hedged findings in this fashion throughout.

These, and changes elsewhere, did not change any conclusions directly, however, one test did cross the significance threshold, namely for genes in proximity to SNPs associated with childhood cancer risk (final paragraph of the Results section). Before we incorporated changes based on your suggestions we saw a barely significant difference in the LOEUF score associated with these 49 genes, however, this cross back over the 0.05 threshold

when we corrected for gene size. This is by no means surprising, as explained in the text (also in the original submission).

2. Number of LoF variants in children vs. in healthy adults

My concern largely persists regarding the analysis shown in Fig 4. I appreciate that the authors explain how they calculate the expected number by extrapolation, but the assumption of a parabolic relationship between the number of unique variants and sample size is poorly justified. Such a relationship is only expected for neutral variants under a constant-sized population but fits poorly for LoF variants in humans due to recent super-exponential population expansion of human populations and the deleteriousness of variants (which leads to enrichment in low frequencies). Consequently, the results shown in Fig 4 are still unreliable. In contrast, the downsampling approach is more trustworthy and should be focused on in the text and figure. Given that the gnomAD dataset does not provide individual-level data, the authors generated an approximate dataset by randomly assigning the observed number of variants to 125,748 rows (each representing an individual). However, this random assignment ignores linkage disequilibrium among variants and hence may not represent the actual distribution of variants among individuals. For best results, I recommend the authors carry out subsampling from actual individual-level data such as the UK10K. Alternatively, the authors need to explicitly describe the methods used in main text and clearly lay out the assumptions and caveats of their approach. I also recommend the authors carry out the second analysis that I suggested (e.g., compare the average number of LoF per individual between adults and children with cancer, by averaging the allele frequencies of all relevant LoF variants observed in each group). Lastly, to make sure there isn't any technical problems with the genome-sequencing or variant-calling in the pediatric cancer studies, it would be helpful to repeat the analysis for a random set of unconstrained genes as a negative control.

Thank you for elaborating this point, and for your recognition of our efforts. We understand your continued concerns and have now reworked the section substantially so as to eliminate the assumptions you highlight entirely. We feel that this improves the clarity of the state of current data on pediatric cancer cohorts greatly. The change was made possible through access to actual down-sampled gnomAD data (v2.1 exomes, n=125,748) which downsamples on individual-data level, thus now accounting for linkage. To compare this data to that reported for pediatric cancer cohorts, we took the actual total data and then also downsampled this to the same precomputed levels used for the gnomAD data (as allowed by cohort-size). Please see revised Figure 4. Removal of these problematic extrapolations required changes in the text, exactly as you pointed out.

The downsampling of the compiled peds cancer data was done iteratively as is now described in the revised methods and as shown in the new version of the code on GitHub. We sincerely hope that you agree that since we are no longer extrapolating or downsampling while blind to linkage, these data are far less problematic. We absolutely agree with you that individual-level access to another large cohort with genomic data, such

as UKbiobank, would strengthen many of our findings, but such expanded analysis are outside the scope of the current work. We hope that it'll serve as a next step for deeper exploration of this topic.

3. Simulation of childhood cancer risk variants

Several key assumptions and parameters need to be clarified. First please clarify the assumption about dominance of the pathogenic variants in the simulation. I presume fully dominant effect is assumed but this needs to be explicitly stated in the text. Second, the population size (e.g., number of individuals) simulated needs to be stated in the text. The authors wrote "an infinite population" in their github code, but this cannot be true, because there is no genetic drift in a truly infinite population, and no variants would ever be lost. Overall, the simulation seems like an overkill to illustrate the point that "even highly incomplete penetrance will lead to mutational constraint". Given complete lethality and full dominance, the penetrance level is equivalent to selection coefficient against the variant. With a finite population size of $N_e=20,000$ (the current estimated for human), a selection coefficient of 0.1% (which is equivalent to 0.1% penetrance under this model) is already considered to be strongly deleterious. Therefore, the simulation results can be largely replaced by simple back-of-envelope calculation/explanation.

We appreciate this commentary. Based on a lengthy discussion, we have elected to remove this supplemental aspect entirely. Its purpose was, as stated in the previous submission, to serve "for illustrative purposes". We felt that it illustrated well how even reduced penetrance of something severe could generate constraint – of course obvious to you, but perhaps less so to many potential readers of this work. The purpose of having an "infinite" population was merely to assure that no inbreeding at any level would occur. Genetic drift is rendered inconsequential as the model only concerns de novo variants; hence, starting with a single carrier, the variant can certainly be lost; regardless of the size of the population that carrier can potentially procreate with.

However, ultimately the removal was done because we agree with you that the simulation was largely superfluous and we also fear that it may create more confusion than clarity. Then, based on the fact that the simulation is inconsequential for our findings and conclusion, we removed it.

4. As I explained in my review last time, although there is evidence that pCPS genes are subjected to strong selective pressure, it is unclear if the selection constraint is fully driven by childhood cancer risk (obviously childhood cancer place selective constraint, but it is questionable whether other subclinical phenotypes also contribute). Therefore, I suggest modifying the title to something like "the evolutionary constraint of genes underlying childhood cancer risk".

We have closely considered this point, and appreciate your suggestion. We were very careful in constructing our title so that it 1) did not make claims about causation and 2) had broad appeal. We agree that your suggestion is technically more precise and would be happy for the work to be published under your suggested title. In the end, the decision is perhaps best left to the editor as we can see scientific merit in both, while she may see other qualities in one vs. the other.

5. Choice of statistical test

It is inappropriate to use t-test to compare LOEUF score, gene length or other metrics of two groups of genes, as these metrics are usually not normally distributed; instead, please use a nonparametric test (e.g., Mann-Witney U test, or K-S test) to compare the distributions. This applies to the numerous places in the paper (e.g., in lines 100-101, 118-119, 121, 123-124...). We are very grateful that you made this observation (which had eluded both us and the other reviewers). You are of course correct and we have rerun all relevant tests. Outcomes and conclusions remain unchanged.

Minor points:

1. “Selective mutational pressure” in the abstract can be replaced by “selective pressure”.

Corrected.

2. The word “put” is unnecessary in line 57.

Corrected.

3. The two sentences in lines 59-61 and 61-63 are redundant and repetitive. Please remove one.

Thank you; revised with deletion of 61-63.

4. Figure 1, the top panel describing the survival bias with WWII bombers is unnecessary; the authors can keep the genome illustration (the lower half of Fig 1A) if wanted. In addition, the label “Likely not constraint [LOELF<35%]” in Fig 1B is incorrect; I believe it should be “Likely not constraint [LOELF>35%]”. In the legend of Fig 1B, LOELF and LOEUF should represent the “lower and upper bounds of the 90% confidence interval” instead of “lower and upper fraction of 90% confidence interval”.

Reviewer #3 also suggested edits to Fig1A, and we have revised it based on your and their suggestions. The “Likely not constraint [LOELF<35%]” is correct, i.e., the lower part of the CI must include the 35% cut-off – the inference required is that LOE is greater than 35%. The figure has been corrected to make this clear.

As the F in LOEUF, LOELF etc. responds to ‘fraction’, this should be included in the written-out definition, but you are right that the “bound” was missing here; and several other places. This has now been corrected throughout so it exactly matches Karczewski et al., 2020, Nature.

5. Please explain how the subset of genes matched to pCPS genes by size was selected. This explanation was in the supplementary material of the previous submission under the subheader “Size-matched gene set”, and we are unsure if you perhaps missed this (which is

perfectly excusable) or whether you found the explanation lacking? We feel that the size-matching approach is sufficiently explained for the purposes of the method section, and for any reader with particular interest, the code, and data to run in full is freely available. A reference to the specific section of the code is made in the relevant section under subheader "Size-matched gene set". If there is a need for deeper methodological description, we would greatly appreciate your specific guidance.

6. In line 224, "be founded" can be replaced with "newly arise".

Corrected.

Reviewer #5 (Remarks to the Author):

I have read the revised version of the manuscript and the authors' comments on the revision. The revision follows the editorial recommendation to remove one of the sections. I consider this has dealt with the most contentious points I originally raised. I appreciate the open-mindedness with which the authors have considered my and the other reviewers' comments. I am satisfied that my concerns have been addressed in the revision.

Thank you very much for contributing your expertise and time, and for guiding us in the revision, which we found greatly improved our work.

Reviewer #3 (Remarks to the Author):

This further revision of the manuscript is significantly improved and the authors have been highly responsive to my prior sets of comments.

Two remaining minor comments:

1. When discussing those genes that don't show restraint, e.g. VHL they note that SDHD does show constraint even though it is imprinted. More recent data suggests that SDHD itself is not imprinted but that the second hit which occurs during tumorigenesis involves a nearby imprinted locus - see for example, Hum Mol Genet. 2016 Sep 1;25(17):3715-3728. doi: 10.1093/hmg/ddw218.

2. Early in the paper, line 111, there is a missing reference.

Reviewer #4 (Remarks to the Author):

I appreciate that the authors are very receptive to feedback and followed the reviewers' suggestions to remove the section on simulations of propagation of de novo mutations and switch to non-parametric statistical tests throughout the manuscript. The revised figures also look much clearer.

The authors have adequately address most of my concerns, with the exception of the downsampling analysis (corresponding to Figure 4). The methods (in the Supplementary Materials) for this analysis are still insufficiently described, and I have a few remaining questions:

1. The authors stated in their response that "The change was made possible through access to actual down-sampled gnomAD data (v2.1 exomes, n=125,748) which downsamples on individual-data level, thus now accounting for linkage." My understanding is that gnomAD does not provide individual-level genotype data except for a few thousand individuals in the 1000G and HGDP projects. Consistent with this, the vcf file (gnomad.exomes.r2.1.1.sites.vcf.bgz) directly accessible from gnomAD website does not contain individual-level data. I hope the authors could clarify how they acquired the individual-level data from gnomAD and what is the actual sample size of the individual-level data.

2. The methods section stated that “To make variation observed in children with cancer comparable to those seen in gnomAD, we used pre-computed downsamples of the gnomAD data (see data availability; see primary code, header: Figure 4 (incl. gnomad data import), chunk: downsampling_import) filtered to variation affecting canonical transcripts.” However, the primary code on github simply imports a precomputed downsampling file “gnomad.v2.1.1.lof_metrics.downsamplings.txt” without explaining where this file is downloaded or how it is generated. I think it is more important to explain how the downsampling is done than how the plot is made in the primary code.

3. How are the expected numbers of LoF variants (grey dots) in Fig 4C computed? Please explain the methods and state the assumptions involved. (These numbers seem to be part of the “gnomad.v2.1.1.lof_metrics.downsamplings.txt” file, but it is again unclear how these numbers are calculated or directly obtained from gnomAD.)

4. In general, I think it is necessary for the authors to clearly describe how the analysis is performed using plain text in the Methods section. The code is a helpful addition that improves reproducibility, but it should not be a substitution of text description of the methods.

Reviewer #3 (Remarks to the Author)

This further revision of the manuscript is significantly improved and the authors have been highly responsive to my prior sets of comments.

Thank you for this message and for recognizing our effort to include your suggestions.

Two remaining minor comments:

1. *When discussing those genes that don't show restraint, e.g. VHL they note that SDHD does show constraint even though it is imprinted. More recent data suggests that SDHD itself is not imprinted but that the second hit which occurs during tumorigenesis involves a nearby imprinted locus - see for example, Hum Mol Genet. 2016 Sep 1;25(17):3715-3728. doi: 10.1093/hmg/ddw218.*

Thank you for bringing this to our attention. We have clarified the text and included a reference to the paper you've listed in order to make this more clear to the reader:

II. 284-8 Revised to: "Intriguingly, the penetrance for variants in ~~the imprinted gene~~ SDHD was twice as high as for other SDHx mutations, and more variants in SDHD (84%) were LoF. Seemingly in accordance with this, SDHD is the only succinate dehydrogenase gene that is likely constrained for LoF mutations (LOE ratio of less than 0.35), even though a phenotype is only present with paternal transmission (**ref to doi: 10.1093/hmg/ddw218**)."

2. *Early in the paper, line 111, there is a missing reference.*

We apologize for this mistake; the placeholder text

"XXXneed_referencing_zotero_down_errorXXX", which erroneously made it into the latest submission has been replaced with intended references to the *NCBI Gene Reviews* for the relevant genes "*PIK3CA, BRAF, RET & PTPN11*".

Reviewer #4 (Remarks to the Author)

I appreciate that the authors are very receptive to feedback and followed the reviewers' suggestions to remove the section on simulations of propagation of de novo mutations and switch to non-parametric statistical tests throughout the manuscript. The revised figures also look much clearer.

Thank you for this kind message; we are highly grateful for your continued willingness to assist us in improving our work and appreciate that you recognize our sincere efforts to address your suggestions.

The authors have adequately address most of my concerns, with the exception of the downsampling analysis (corresponding to Figure 4). The methods (in the Supplementary Materials) for this analysis are still insufficiently described, and I have a few remaining questions:

1. The authors stated in their response that “The change was made possible through access to actual down-sampled gnomAD data (v2.1 exomes, n=125,748) which downsamples on individual-data level, thus now accounting for linkage.” My understanding is that gnomAD does not provide individual-level genotype data except for a few thousand individuals in the 1000G and HGDP projects. Consistent with this, the vcf file (gnomad.exomes.r2.1.1.sites.vcf.bgz) directly accessible from gnomAD website does not contain individual-level data. I hope the authors could clarify how they acquired the individual-level data from gnomAD and what is the actual sample size of the individual-level data.

Thank you for these comments. There is a miscommunication, surely due to a lack of clarity on our part, which addresses the majority of your very valid concerns. The reference to the external gnomAD downsampling data was provided under the journal's required header "Data Availability", however, the body text was misleadingly colored as a hyperlink; here is the direct text from the submission with the hyperlinking fixed: "[...] and associated downsampling data from https://storage.googleapis.com/gcp-publicdata-gnomad/release/2.1.1/constraint/gnomad.v2.1.1.lof_metrics.downsamplings.txt.bgz". The link leads directly to the data; it is also available here from gnomAD's main download page. The data contains summary stats on downsampled gnomAD data. The downsampling was done internally at the gnomAD project (which of course has individual-level data) and then released in the data format above.

To fully clarify this we have: (1) added a reference to the direct data download in the code itself; (2) added a reference to the Data availability in the relevant manuscript text and figure; (3) added a paragraph explaining the origin of the data, including the computation of LOEUF scores; with reference to Karczewski et al., Nature, 2020 (see response to point #4).

2. The methods section stated that “To make variation observed in children with cancer comparable to those seen in gnomAD, we used pre-computed downsamples of the gnomAD data (see data availability; see primary code, header: Figure 4 (incl. gnomad data import), chunk: downsampling_import) filtered to variation affecting canonical transcripts.” However, the primary code on github simply imports a precomputed downsampling file “gnomad.v2.1.1.lof_metrics.downsamplings.txt” without explaining where this file is downloaded or how it is generated. I think it is more important to explain how the downsampling is done than how the plot is made in the primary code. We fully agree. Your concerns are in line with the points raised above (please confer response to point #1). We agree with your point and have expanded to the methods to explain the downsampling in text (corresponding exactly to the provided code).

3. How are the expected numbers of LoF variants (grey dots) in Fig 4C computed? Please explain the methods and state the assumptions involved. (These numbers seem

to be part of the “gnomad.v2.1.1.lof_metrics.downsamplings.txt” file, but it is again unclear how these numbers are calculated or directly obtained from gnomAD.) Please confer with our response to point #1 and especially point #4, which should clarify this fully.

4. In general, I think it is necessary for the authors to clearly describe how the analysis is performed using plain text in the Methods section. The code is a helpful addition that improves reproducibility, but it should not be a substitution of text description of the methods.

We agree and have now extended the methods with this in mind. This text also addresses your other 3 points (revised additions are in **bold**):

" To make variation observed in children with cancer comparable to those seen in gnomAD, we used precomputed downsamples of the gnomAD data (see **under header 'Data Availability' in the main manuscript text**; see primary code, header: Figure 4 (incl. gnomad data import), chunk: downsampling_import) filtered to variation affecting canonical transcripts. Importing this data (

https://storage.googleapis.com/gcp-publicdata--gnomad/release/2.1.1/constraint/gnomad.v2.1.1.lof_metrics.downsamplings.txt.gz), we generated plots comparing the observed number of LoF variation in the two

groups by downsampling the combined germline data reports from pediatric pancancer studies (downsampling was done, as allowed by sample size, to the same levels as those precomputed in the gnomAD data). **A total of 38 possible downsampling steps were used [10, 20, 50, 100, 200, 500, 1000, 2000, 3070, 5000, 5040, 8128, 9197, 10000, 10824, 15000, 15308, 17296, 20000, 25000, 30000, 35000, 40000, 45000, 50000, 55000, 56885, 60000, 65000, 70000, 75000, 80000, 85000, 90000, 95000, 100000, 110000, 120000].** In the gnomAD dataset, at each of these downsampling steps the observed number of distinct LoF variants were pre-computed, as were the *expected* number of distinct LoF variants, calculated using the same methods employed for the gnomAD-wide calculations, as detailed by Karczewski et al. (Nature, 2020 ref).

In our downsampling of the pediatric pancancer cohort data we employed an iterative for-loop (see primary code, header: Figure 4 (incl. gnomad data import), chunk: downsampling_for_pancan). This meant that the for-loop began by counting all distinct LoF variants (as defined below) across all available studies (variable by number of studies with the given gene included on panel (see Supplementary Data 16-26)). For genes included in all 11 pediatric pancancer studies, the total number of childhood cancer cases with data was 4,574 (9,148 alleles). Each unique patient was labelled either with their specific variant or as having no variant detected. From this set, using the sample_n function in the R package dplyr, a random sample, corresponding to the nearest lower downsample step (n

= 3,070 in the example), was extracted. LoF variants in the extracted random sample were counted, and then, a random sample corresponding to the next downsample step (n = 2,000 in the example) was taken from the previous step (n = 3,070 in the example). This process ran iteratively through all downsampling steps (9 steps [10, 20, 50, 100, 200, 500, 1000, 2000, 3070] in the example)."

Also we made these updates to the legend of Figure 4 (updates in **bold**):

"Figure 4: [...] C: Log-scaled point graphs showing number of distinct LoF variants at various sample sizes, with colors representing gnomAD data, pediatric pan-cancer data or expected number as indicated. **For gnomAD data and pediatric pan-cancer data, the dots with the highest x-axis value of each plot correspond to actually observed LoF variants in the full data, with other dots representing number of observed LoF variants at 38 downsampling steps (precomputed for gnomAD with pediatric pan-cancer internally computed to match, see Methods and Data Availability).** The expected number of LoF variants at each downsampling step were precomputed as detailed by Karczewski et al. (Nature 2020 ref) and is **publicly available from gnomAD (see Data Availability)** *BAP1 and CDKN2A were reported in 10 pediatric pan-cancer studies, but were not included in this figure due to figure size/readability. Graphs for all 11 genes not shown here are in Supplementary Figure 3 . 100K, 100,000 individuals, LOE, LoF observed vs. expected ratio, 90%CI, 90% confidence interval."